# ADAPTIVE SCALING OF POLICY CONSTRAINTS FOR OFFLINE REINFORCEMENT LEARNING

**Jing Tan**
University of Science and Technology Beijing
Beijing, China
Hong Kong University of Science and Technology (Guangzhou)
Guangzhou, China

**Xiaorui Li, Chao Yao, Xiaojuan Ban & Zhaolin Yuan**[*]
University of Science and Technology Beijing
Beijing, China
yuanzhaolin@ustb.edu.cn

**Yuetong Fang & Renjing Xu**[*]
Hong Kong University of Science and Technology (Guangzhou)
Guangzhou, China
renjingxu@hkust-gz.edu.cn

## ABSTRACT

Offline reinforcement learning (RL) enables learning effective policies from fixed datasets without any environment interaction. Existing methods typically employ policy constraints to mitigate the distribution shift encountered during offline RL training. However, because the scale of the constraints varies across tasks and datasets of differing quality, existing methods must meticulously tune hyperparameters to match each dataset, which is time-consuming and often impractical. To bridge this gap, we propose Adaptive Scaling of Policy Constraints (ASPC), a second-order differentiable framework that automatically adjusts the scale of policy constraints during training. We theoretically analyze its performance improvement guarantee. In experiments on 39 datasets across four D4RL domains, ASPC using a single hyperparameter configuration outperforms other adaptive constraint methods and state-of-the-art offline RL algorithms that require per-dataset tuning, achieving an average 35% improvement in normalized performance over the baseline. Moreover, ASPC consistently yields additional gains when integrated with a variety of existing offline RL algorithms, demonstrating its broad generality.

## 1 INTRODUCTION

Offline reinforcement learning (RL) learns a policy exclusively from a fixed, pre-collected dataset without further interactions with the environment Levine et al. (2020). This characteristic is particularly crucial in real-world applications such as autonomous driving El Sallab et al. (2017); Kendall et al. (2019), healthcare Prasad et al. (2017); Wang et al. (2018), industry Zhan et al. (2022); Yuan et al. (2024), and other tasks, where interacting with the environment can be expensive and risky.

Despite the potential advantages, a critical challenge in offline RL is the distribution shift Levine et al. (2020) between the offline data and the training policies, often leading to suboptimal or even invalid policy updates. Many methods have been proposed to mitigate the adverse effects of the distribution shift. A common strategy is to impose explicit or implicit policy constraints Fujimoto et al. (2019); Kumar et al. (2020); Fujimoto & Gu (2021); Kostrikov et al. (2022), ensuring that the learned policy remains close to the behavior policy used to collect the dataset. By imposing constraints on policy updates, these methods can effectively mitigate the extrapolation error of the Q value Fujimoto et al. (2019) induced by the distribution shift while offering certain performance guarantees.

A central but often overlooked issue in policy constraint methods is the choice of the constraint scale, which crucially governs the balance between the RL objective and the behavior cloning (BC) term. Existing approaches fall into two categories. First, methods that rely on dataset-specific hyperparameter tuning can achieve strong results, but their performance collapses once a single

---

[*]Corresponding authors

configuration is applied across tasks or datasets of varying quality, as shown in Figure 1(b). Second, adaptive variants with fixed hyperparameters Peng et al. (2023); Yang et al. (2024) alleviate tuning costs, yet they only reweight actions locally and neglect the global trade-off scale, leaving a significant gap to carefully tuned baselines. In practical offline RL, where extensive tuning is prohibitively expensive or even infeasible, the pressing challenge is how to achieve robust performance with a single hyperparameter configuration across diverse datasets.

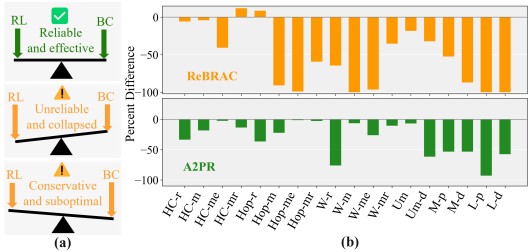

Figure 1: (a) The RL–BC trade-off in offline RL. ASPC dynamically balances RL and BC, yielding a reliable and effective policy (left). Existing methods fail to properly calibrate this trade-off, resulting in suboptimal or collapsed policies (middle and right). (b) Percent difference in performance for ReBRAC and A2PR under a single hyperparameter setting across all datasets. HC = HalfCheetah, Hop = Hopper, W = Walker, r = random, m = medium, mr = medium-replay, me = medium-expert, Um = umaze, M = medium, L = large, d = diverse, p = play.

To enable a single hyperparameter configuration to match or exceed the performance of finely tuned methods across datasets of varying quality and tasks, we propose an adaptive scaling of policy constraints (ASPC) approach that dynamically adjusts the constraint scale during training. The intuition of this method is shown in Figure 1(a). Our approach leverages a second-order differentiable optimization framework Finn et al. (2017) to balance the goals of RL and BC. Specifically, we parameterize the scale factor $\alpha$ as a learnable parameter that balances the RL objective $\mathcal{L}_{\mathrm{RL}}$ and the BC objective $\mathcal{L}_{\mathrm{BC}}$ in TD3+BC Fujimoto & Gu (2021). The combined objective $\mathcal{L}$ is given by

$$\mathcal{L} = \alpha \mathcal{L}_{\mathrm{RL}} + \mathcal{L}_{\mathrm{BC}}, \tag{1}$$

For the full definitions of $\alpha$, refer to equation 3. During training, $\alpha$ is dynamically adjusted by constraining the rate of change of the Q-value and the BC loss, enabling the algorithm to discover a more stable learning path and exhibit remarkable adaptability across tasks and datasets.

We theoretically analyze the performance improvement guarantee of ASPC and extensively evaluate it on the D4RL benchmark Levine et al. (2020). Our empirical results demonstrate that ASPC outperforms other state-of-the-art offline RL algorithms that depend on meticulously tuned hyper-parameters for each dataset, while adding only minimal computational overhead to the original TD3+BC backbone. In addition, ASPC improves a variety of offline RL algorithms beyond TD3+BC, further indicating its generality and broad applicability.

## 2 RELATED WORKS

### 2.1 OFFLINE RL

Offline RL aims to learn policies purely from static datasets and suffers from distribution shift between the behavior policy and the learned policy, leading to value overestimation and policy collapse. Existing approaches address this challenge from several perspectives. Policy constraint methods explicitly Fujimoto et al. (2019); Fujimoto & Gu (2021) or implicitly Kumar et al. (2020); Kostrikov et al. (2022) regularize the learned policy toward the behavior distribution. Uncertainty-aware approaches penalize actions with high epistemic or aleatoric uncertainty An et al. (2021); Bai et al. (2022); Zhang et al. (2023). Sequence modeling methods reformulate RL as conditional trajectory modeling using transformers Chen et al. (2021); Janner et al. (2021). Among these, policy constraint methods have emerged as the most direct and widely adopted solution, but their effectiveness crucially depends on properly scaling the constraint. This motivates our focus on developing an adaptive scaling mechanism that eliminates the need for per-dataset tuning while retaining robustness across diverse offline RL benchmarks.

## 2.2 ADAPTIVE POLICY CONSTRAINTS

Balancing the RL objective against BC is central to offline RL, and the strength of this constraint critically affects both stability and performance. Recent work has explored adaptive ways to tune this balance. Trajectory- or sample-weighting methods such as AW Hong et al. (2023), wPC Peng et al. (2023), and OAP Yang et al. (2023) reweight transitions or actions based on estimated value or expert preference, thereby adjusting constraint strength locally. Other approaches introduce auxiliary models to refine constraint scaling, for example PRDC Ran et al. (2023), GORL Yang et al. (2024), A2PR Liu et al. (2024), and IEPC Liu & Hofert (2024). Despite these advances, current approaches either rely on per-dataset hyperparameter tuning for optimal performance, or apply a fixed configuration that yields only limited gains across domains. Our ASPC method addresses this gap by dynamically adjusting the constraint scale during training, enabling robust performance across diverse datasets with a single hyperparameter configuration.

## 3 PRELIMINARIES

RL problems are formulated as a Markov decision process (MDP), described by the tuple $(S, A, P, R, \gamma)$. The set of states is $S$, the set of actions is $A$, the transition probability function is $P(s'|s, a)$, the reward function is $R(s, a)$, and $\gamma \in [0, 1)$ is the discount factor. The objective is to find a policy $\pi : S \to A$ that maximizes the expected discounted return. This objective is equivalently expressed as maximizing the Q-value $Q^\pi(s, a)$ under $\pi$, given by:

$$Q^\pi(s, a) = \mathbb{E}_\pi \left[ \sum_{t=0}^{\infty} \gamma^t R(s_t, a_t) \,\middle|\, s_0 = s, a_0 = a \right], \tag{2}$$

where $s_t$ and $a_t$ represent the state and action at time $t$. In practice, RL algorithms update Q-values using the Bellman equation as an iterative rule, seeking to converge to the optimal policy $\pi^*$.

A central challenge for offline RL is the distribution shift. When a state–action pair $(s, a)$ lies outside the dataset $\mathcal{D}$, directly optimizing the Q–function may cause severe over-estimation. One remedy is to constrain the target policy $\pi$ to stay close to the behaviour policy $\pi_\beta$. TD3+BC Fujimoto & Gu (2021) does so by solving:

$$\pi = \arg \max_\pi \; \mathbb{E}_{(s,a)\sim\mathcal{D}} \Big[ \lambda \underbrace{Q(s, \pi(s))}_{\text{RL}} - \underbrace{(\pi(s) - a)^2}_{\text{BC}} \Big], \quad \lambda = \frac{\alpha}{\frac{1}{N} \sum_i |Q(s_i, a_i)|}. \tag{3}$$

normalizes the RL term to the scale of the BC loss. In vanilla TD3+BC, $\alpha$ is a fixed constant. Instead of keeping the scale factor $\alpha$ static, we update it throughout training.

## 4 METHOD

We now present the ASPC algorithm in detail. We begin by introducing its core framework, a second-order differentiable optimization that adaptively balances the RL and BC objectives (Section 4.1). We then provide a theoretical analysis (Section 4.2), which explains the role of the mutual constraint term and establishes single-step and long-term performance guarantees. Finally, we describe a practical instantiation of ASPC built on TD3+BC (Section 4.3), which enables its application to standard offline RL benchmarks.

### 4.1 ADAPTIVE SCALING OF POLICY CONSTRAINTS

To adaptively adjust the relative scaling between the RL and BC objectives, ASPC adopts a meta-learning approach Finn et al. (2017); Franceschi et al. (2018). It converts the scale factor $\alpha$ in equation 3 into a learnable parameter and optimizes it dynamically via bilevel training, utilizing inner updates and outer updates to maximize RL exploration near the behavior policy.

**Inner Update** To optimize the policy under offline data, we define the inner objective as

$$\mathcal{L}_{\text{inner}}(\theta; \alpha) = \mathbb{E}_{(s,a)\sim\mathcal{D}} \Big[ -\lambda(\alpha) \, Q\big(s, \pi_\theta(s)\big) + \|\pi_\theta(s) - a\|^2 \Big], \tag{4}$$

where $\lambda(\alpha) = \alpha/\mathbb{E}_{s\sim\mathcal{D}}[|Q(s,\pi_\theta(s))|]$. The inner update is then obtained via a gradient descent step with learning rate $\eta_\theta$:

$$\tilde{\theta}(\alpha) \;=\; \theta \;-\; \eta_\theta\,\nabla_\theta\mathcal{L}_{\text{inner}}(\theta;\alpha), \tag{5}$$

and $\tilde{\theta}(\alpha)$ denotes the updated policy parameters after one inner step.

**Outer Update** While the inner update optimizes the policy parameters $\theta$ for a given scale $\alpha$, the outer update is responsible for adjusting $\alpha$ itself so as to dynamically balance the RL and BC objectives. The outer loss is composed of three coordinated components. $\mathcal{L}_1$ mirrors TD3 + BC and steers $\alpha$ toward a better balance between RL and BC. $\mathcal{L}_2$ penalizes abrupt increases in the expected Q-value, while $\mathcal{L}_3$ constrains large shifts in the BC loss. Together, $\mathcal{L}_2$ and $\mathcal{L}_3$ adaptively regulate the step prescribed by $\mathcal{L}_1$, preventing either RL or BC from dominating and thereby stabilizing training. Formally, we write:

$$\mathcal{L}_1 = -\alpha\,\frac{\mathbb{E}_{s\sim\mathcal{D}}\big[Q\big(s,\pi_{\tilde{\theta}}(s)\big)\big]}{\mathbb{E}_{s\sim\mathcal{D}}\big[\big|Q\big(s,\pi_{\tilde{\theta}}(s)\big)\big|\big]} \;+\; \mathbb{E}_{(s,a)\sim\mathcal{D}}\Big[\big\|\pi_{\tilde{\theta}}(s) - a\big\|^2\Big], \tag{6}$$

$$\mathcal{L}_2 = \Big(\mathbb{E}_{s\sim\mathcal{D}}\big[Q\big(s,\pi_{\tilde{\theta}}(s)\big)\big] - \mathbb{E}_{s\sim\mathcal{D}}\big[Q\big(s,\pi_\theta(s)\big)\big]\Big)^2, \tag{7}$$

$$\mathcal{L}_3 = \big(\mathcal{L}_2.detach\big)\Big(\sup_{(s,a)\in\mathcal{D}}\|\pi_\theta(s) - a\|^2\Big)\Big(\sup_{(s,a)\in\mathcal{D}}\big|\|\pi_{\tilde{\theta}}(s) - a\|^2 - \|\pi_\theta(s) - a\|^2\big|\Big), \tag{8}$$

The outer objective is

$$\mathcal{L}_{\text{outer}}\big(\tilde{\theta}(\alpha)\big) = \mathcal{L}_1 + \mathcal{L}_2 + \mathcal{L}_3. \tag{9}$$

Here, $\pi_\theta$ and $\pi_{\tilde{\theta}}$ denote the policies before and after the inner update, respectively. $.detach$ indicates stopping gradients. While $\mathcal{L}_1$ and $\mathcal{L}_2$ are relatively standard, the design of $\mathcal{L}_3$ requires clarification. Theoretically, its form follows directly from Theorem 4.4, with details in Appendix A.3. Intuitively, $\mathcal{L}_3$ combines three factors: the rate of change in Q-values, the upper bound of the BC loss, and the variation in BC loss across iterations. Large Q-value fluctuations or a high BC-loss bound signal rapid policy change or significant deviation from the behavior policy. In such cases, strengthening the penalty on BC variation helps suppress distributional shift and stabilize training, consistent with our intuition. To update $\alpha$, we treat the inner update parameters $\tilde{\theta}(\alpha)$ as an implicit function of $\alpha$ and use second-order derivatives. Lets $\eta_\alpha$ be the learning rate of $\alpha$. The gradient-descent step is

$$\alpha \leftarrow \alpha - \eta_\alpha\bigg(\frac{\partial\mathcal{L}_{\text{outer}}\big(\tilde{\theta}(\alpha)\big)}{\partial\tilde{\theta}}\,\frac{\partial\tilde{\theta}(\alpha)}{\partial\alpha}\bigg), \tag{10}$$

## 4.2 THEORETICAL ANALYSIS

We now analyze the theoretical properties of ASPC. We show that the outer objective ensures stable updates and reduces the gap to the optimal policy.

**Assumption 4.1.** The critic $Q(s,a)$ and the transition kernel $P(\cdot \mid s,a)$ are Lipschitz continuous with respect to the action variable. That is, there exist constants $L_Q, L_P > 0$, independent of $s$, such that for all $s \in \mathcal{S}$ and all $a_1, a_2 \in \mathcal{A}$,

$$\|Q(s,a_1) - Q(s,a_2)\| \le L_Q\|a_1 - a_2\|, \|P(\cdot \mid s,a_1) - P(\cdot \mid s,a_2)\|_{\text{TV}} \le L_P\|a_1 - a_2\|. \tag{11}$$

**Proposition 4.2** (Mutual constraints between $\Delta L_{BC}$ and $(\Delta Q)^2$). *Under Assumption 4.1, the change in BC loss ($\Delta L_{BC}$) and the squared change in Q-values ($(\Delta Q)^2$) mutually constrain each other: $(\Delta Q)^2$ provides a lower bound on $\Delta L_{BC}$, while $\Delta L_{BC}$ provides an upper bound on $(\Delta Q)^2$.*

This result shows that the two penalties in equation 7 and equation 8 are inherently coupled rather than independent. It explains why in practice some tasks succeed with only one of them, while others require both for stable training (see Section 5.5). The detailed proof is provided in Appendix A.1.

**Proposition 4.3** (Single-step performance lower bound). *For the update step from $\pi_t$ to $\pi_{t+1}$, the performance improvement admits the following lower bound:*

$$J(\pi_{t+1}) - J(\pi_t) \;\ge\; \frac{1}{1-\gamma}\Big(\Delta Q - \Phi(\Delta L_\infty^{BC}, c_\infty^2)\Big), \tag{12}$$

*where $\Phi(\Delta L_\infty^{BC}, c_\infty^2)$ is a nonnegative function depending on the BC-loss variation upper bound $\Delta L_\infty^{BC}$ and the BC-loss upper bound $c_\infty^2$.*

This proposition serves as the theoretical basis for Theorem 4.4. It also directly motivates the design of the penalty term $\mathcal{L}_3$ (equation 8), whose form is derived from bounding $\Phi(\Delta L_\infty^{BC}, c_\infty^2)$. The detailed derivation of $\Phi$ is deferred to Appendix A.2.

**Theorem 4.4** (Single-step performance condition for ASPC). *An idealized ASPC update that satisfies the condition $\Delta Q \geq \Phi$ leads to a non-decreasing policy performance: $J(\pi_{t+1}) - J(\pi_t) \geq 0$.*

ASPC employs a smooth relaxation of this condition via the outer objective, which is designed to guide updates toward this provably stable regime. The detailed proof is given in Appendix A.3.

**Theorem 4.5** (Performance gap to optimal). *With Theorem 4.4, after $T$ iterations when the single-step gain vanishes ($\delta_T = 0$), the gap to the optimal policy satisfies:*

$$J(\pi^*) - J(\pi_T) \ \leq \ \Psi(\varepsilon_\beta) - T\,\delta_{\min}, \tag{13}$$

*where $\Psi(\varepsilon_\beta)$ is a function of the mismatch $\varepsilon_\beta$ between the behavior policy and the optimal policy, and $\delta_{\min}$ denotes the minimal single-step improvement before convergence.*

This theorem shows that ASPC progressively reduces the suboptimality gap until convergence, where the remaining gap is controlled by $\Psi(\varepsilon_\beta)$. The full derivation of $\Psi(\varepsilon_\beta)$ is given in Appendix A.4.

---

**Algorithm 1** Adaptive Scaling of Policy Constraints
---
**Initialize:** critic and actor networks, scale factor $\alpha$, replay buffer $\mathcal{D}$, update intervals $k_\pi$, $k_\alpha$.

1: **for** $i = 1$ to $N$ **do**
2:   **Critic update:**
3:   Sample minibatch from $D$; Compute TD targets and update critic networks;
4:   **if** $i \bmod k_\pi = 0$ **then**
5:     **Actor update (inner):**
6:     Compute $\mathcal{L}_{\text{inner}}(\theta; \alpha)$ by equation 4; Compute $\tilde{\theta}(\alpha)$ by equation 5;
7:     Update actor networks;
8:     **if** $i \bmod (k_\pi \cdot k_\alpha) = 0$ **then**
9:       $\alpha$ **update (outer):**
10:      Compute $\mathcal{L}_{\text{outer}}(\tilde{\theta}(\alpha))$ by equation 9; Update $\alpha$ via equation 10;
11:     **end if**
12:    Soft update critic and actor networks;
13:  **end if**
14: **end for**

---

### 4.3 Implementation on TD3+BC

To make ASPC practical, we instantiate it on top of the TD3+BC backbone with only two modifications: (i) a redesigned critic network, and (ii) a learnable scale factor $\alpha$. All other network components and hyperparameters remain unchanged. See Appendix B.2 for a full specification.

Recent studies show that deeper critics Kumar et al. (2022); Lee et al. (2022) and the insertion of LayerNorm between layers Nikulin et al. (2023); Ball et al. (2023); Tarasov et al. (2024a) can mitigate Q-value over-estimation and improve stability. Following this evidence, we extend the TD3+BC critic from two to three hidden layers and insert a LayerNorm after each layer. An ablation of this choice is provided in Section 5.5.

Algorithm 1 lists the ASPC procedure. Blue highlights indicate lines that differ from the TD3+BC backbone. Although second-order gradients increase cost, we set the $\alpha$-update interval $k_\alpha$ far longer than the actor-update interval $k_\pi$, which maintains performance while sharply reducing runtime. Section 5.4 analyses this trade-off in detail.

## 5 Experiments

In this section we evaluate ASPC on the D4RL benchmark. Section 5.1 compares ASPC with strong baselines to demonstrate its adaptability and overall effectiveness. Section 5.2 analyzes the learning curves of $\alpha$ during training, further illustrating ASPC's adaptive behaviour. Section 5.3 investigates

Table 1: Average normalized score over the final evaluation across four random seeds. The best performance in each dataset is highlighted in **bold**, while the second-best performance is indicated with an underline. Blue shading indicates methods with top domain average performance. The symbol $\pm$ denotes the standard deviation. ✓denotes fixed hyperparameters, whereas ✗denotes dataset-specific ones. *To ensure fairness, TD3+BC and wPC employ the robust critic described in Section 5.5.

| | Task Name | TD3+BC*(✓) | A2PR(✓) | IQL(✗) | wPC*(✓) | ReBRAC(✗) | ASPC (Ours)(✓) |
|---|---|---|---|---|---|---|---|
| HalfCheetah | Random | 10.6 ± 0.7 | 21.1 ± 0.8 | 19.5 ± 0.8 | 18.8 ± 0.7 | **29.5** ± 1.5 | 20.8 ± 0.9 |
| | Medium | 49.6 ± 0.2 | 56.1 ± 0.3 | 50.0 ± 0.2 | 54.8 ± 0.2 | **65.6** ± 1.0 | 58.7 ± 0.4 |
| | Expert | 100.4 ± 0.4 | 99.9 ± 3.2 | 95.5 ± 2.1 | 103.8 ± 2.4 | **105.9** ± 1.7 | 105.1 ± 1.2 |
| | Medium-Expert | 97.9 ± 1.6 | 95.9 ± 6.0 | 92.7 ± 2.8 | 98.9 ± 8.5 | **101.1** ± 5.2 | 99.9 ± 1.2 |
| | Medium-Replay | 45.8 ± 0.2 | 49.0 ± 0.4 | 42.1 ± 3.6 | 48.1 ± 0.2 | **51.0** ± 0.8 | 50.6 ± 0.5 |
| | Full-Replay | 74.5 ± 1.6 | 79.5 ± 1.5 | 75.0 ± 0.7 | 76.7 ± 2.3 | **82.1** ± 1.1 | 79.3 ± 0.9 |
| Hopper | Random | 8.6 ± 0.2 | **20.1** ± 11.6 | 10.1 ± 5.9 | 8.5 ± 1.4 | 8.1 ± 2.4 | 9.4 ± 1.5 |
| | Medium | 62.0 ± 3.0 | 78.3 ± 4.4 | 65.2 ± 4.2 | 81.8 ± 9.8 | **102.0** ± 1.0 | 92.7 ± 7.2 |
| | Expert | 108.2 ± 4.2 | 83.9 ± 6.0 | 108.8 ± 3.1 | 79.1 ± 26.6 | 100.1 ± 8.3 | **112.3** ± 0.4 |
| | Medium-Expert | 103.3 ± 9.2 | 110.8 ± 2.6 | 85.5 ± 29.7 | 109.1 ± 4.5 | 107.0 ± 6.4 | **111.0** ± 2.1 |
| | Medium-Replay | 47.4 ± 35.4 | 98.9 ± 2.0 | 89.6 ± 13.2 | 100.8 ± 0.7 | 98.1 ± 5.3 | **101.3** ± 0.6 |
| | Full-Replay | 90.3 ± 22.9 | 97.1 ± 17.8 | 104.4 ± 10.8 | 105.6 ± 0.6 | 107.1 ± 0.4 | **107.2** ± 0.5 |
| Walker2d | Random | 5.9 ± 3.5 | 1.2 ± 1.5 | 11.3 ± 7.0 | 12.5 ± 10.6 | **18.4** ± 4.5 | 15.6 ± 6.4 |
| | Medium | 62.0 ± 3.0 | 84.2 ± 4.7 | 80.7 ± 3.4 | 89.6 ± 0.3 | 82.5 ± 3.6 | **92.4** ± 5.4 |
| | Expert | 108.2 ± 4.2 | 84.8 ± 49.0 | 96.9 ± 32.3 | 111.5 ± 0.1 | **112.3** ± 0.2 | 110.8 ± 0.1 |
| | Medium-Expert | 103.3 ± 9.2 | 88.2 ± 40.7 | **112.1** ± 0.5 | 110.1 ± 0.5 | 111.6 ± 0.3 | 111.1 ± 0.3 |
| | Medium-Replay | 76.6 ± 12.7 | 84.5 ± 12.3 | 75.4 ± 9.3 | 93.4 ± 3.0 | 77.3 ± 7.9 | **97.6** ± 0.5 |
| | Full-Replay | 88.3 ± 11.7 | **102.5** ± 0.0 | 97.5 ± 1.4 | 99.5 ± 0.5 | 102.2 ± 1.7 | 102.1 ± 0.2 |
| | **MuJoCo Avg** | 70.7 | 74.2 | 72.9 | 77.8 | 81.2 | 82.1 |
| Maze2d | Umaze | 34.5 ± 13.9 | 102.5 ± 6.3 | -8.9 ± 6.1 | 73.1 ± 13.8 | 106.8 ± 22.1 | **128.1** ± 31.8 |
| | Medium | 63.3 ± 63.3 | 90.4 ± 29.6 | 34.8 ± 2.7 | 87.4 ± 48.7 | 105.1 ± 31.6 | **117.8** ± 17.3 |
| | Large | 108.9 ± 43.6 | 177.7 ± 34.2 | 61.7 ± 3.5 | 123.3 ± 70.5 | 78.3 ± 61.7 | **195.8** ± 31.3 |
| | **Maze2d Avg** | 68.9 | 123.53 | 46.2 | 94.6 | 96.7 | 147.2 |
| AntMaze | Umaze | **100.0** ± 0.0 | 92.5 ± 8.3 | 83.3 ± 4.5 | 97.5 ± 5.0 | 97.8 ± 1.0 | 92.5 ± 5.0 |
| | Umaze-Diverse | 87.5 ± 12.5 | 32.5 ± 34.9 | 70.6 ± 3.7 | 75.0 ± 20.8 | 88.3 ± 13.0 | **92.5** ± 9.5 |
| | Medium-Play | 7.5 ± 9.5 | 40.0 ± 7.1 | 64.6 ± 4.9 | **85.0** ± 5.7 | 84.0 ± 4.2 | **85.0** ± 12.9 |
| | Medium-Diverse | 12.5 ± 12.5 | 40.0 ± 25.5 | 61.7 ± 6.1 | **85.0** ± 12.9 | 76.3 ± 13.5 | 70.0 ± 11.5 |
| | Large-Play | 2.5 ± 5.0 | 5.0 ± 8.7 | 42.5 ± 6.5 | **65.0** ± 19.1 | 60.4 ± 26.1 | 55.0 ± 5.7 |
| | Large-Diverse | 2.5 ± 5.0 | 22.5 ± 14.8 | 27.6 ± 7.8 | **65.0** ± 10.0 | 54.4 ± 25.1 | 52.5 ± 18.9 |
| | **AntMaze Avg** | 35.4 | 38.75 | 58.3 | 78.7 | 76.8 | 74.5 |
| Pen | Human | 53.8 ± 15.7 | -2.1 ± 0.0 | 81.5 ± 17.5 | 39.9 ± 12.8 | **103.5** ± 14.1 | 81.1 ± 8.1 |
| | Cloned | 71.7 ± 21.5 | 6.5 ± 6.0 | 77.2 ± 17.7 | 34.6 ± 11.3 | **91.8** ± 21.7 | 87.2 ± 4.2 |
| | Expert | 126.6 ± 24.8 | 51.5 ± 38.4 | 133.6 ± 16.0 | 141.8 ± 11.8 | **154.1** ± 5.4 | 141.2 ± 9.4 |
| Door | Human | 0.0 ± 0.0 | -0.2 ± 0.0 | **3.1** ± 2.0 | -0.2 ± 0.0 | 0.0 ± 0.0 | 0.0 ± 0.0 |
| | Cloned | 0.0 ± 0.0 | -0.3 ± 0.0 | 0.8 ± 1.0 | 0.0 ± 0.0 | **1.1** ± 2.6 | 0.0 ± 0.0 |
| | Expert | 81.6 ± 16.3 | -0.3 ± 0.0 | 105.3 ± 2.8 | 51.4 ± 55.3 | 104.6 ± 2.4 | **105.6** ± 0.4 |
| Hammer | Human | 0.0 ± 0.0 | 1.1 ± 0.4 | **2.5** ± 1.9 | 0.0 ± 0.0 | 0.2 ± 0.2 | 2.2 ± 3.2 |
| | Cloned | 0.1 ± 0.0 | 0.3 ± 0.0 | 1.1 ± 0.5 | 0.1 ± 0.1 | 6.7 ± 3.7 | **12.0** ± 9.1 |
| | Expert | 132.8 ± 0.4 | 0.3 ± 0.1 | 129.6 ± 71.5 | 57.6 ± 0.1 | **133.8** ± 0.7 | 128.6 ± 0.4 |
| Relocate | Human | 0.0 ± 0.0 | -0.3 ± 0.0 | 0.1 ± 0.1 | 0.1 ± 0.0 | 0.0 ± 0.0 | **0.1** ± 0.2 |
| | Cloned | 0.0 ± 0.0 | -0.3 ± 0.0 | 0.2 ± 0.4 | 0.1 ± 0.0 | **0.9** ± 1.6 | 0.0 ± 0.0 |
| | Expert | 90.6 ± 18.2 | -0.3 ± 0.0 | 106.5 ± 2.5 | 6.7 ± 4.6 | 106.6 ± 3.2 | **111.2** ± 2.4 |
| | **Adroit Avg** | 46.4 | 4.65 | 53.4 | 28.8 | 58.6 | 55.7 |
| | **Total Avg** | 57.7 | 51.2 | 62.6 | 64.2 | 74.8 | 77.9 |

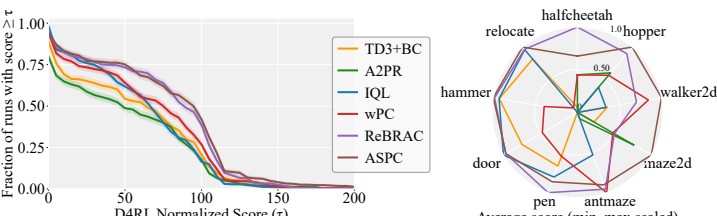

Figure 2: Left: performance profiles on 39 datasets of D4RL. Right: radar chart of the mean performance across the nine tasks.

the necessity of dynamically adjusting $\alpha$. Section 5.4 reports runtime results to highlight the efficiency of ASPC. Section 5.5 presents ablation studies on the key components of ASPC. Section 5.6 provides results on integrating the ideas of ASPC with other methods, and Section 5.7 presents the performance of ASPC on the OGBench benchmark.

## 5.1 COMPARATIVE PERFORMANCE ON BENCHMARK

We evaluate ASPC on 39 datasets spanning four D4RL domains Levine et al. (2020): MuJoCo (v2), AntMaze (v2), Maze2d (v1), and Adroit (v1). Our baselines include TD3+BC Fujimoto & Gu (2021) and IQL Kostrikov et al. (2022) as standard policy-constraint methods. wPC Peng et al. (2023) and A2PR Liu et al. (2024) are state-of-the-art (SOTA) adaptive policy constraint methods built on TD3+BC. ReBRAC Tarasov et al. (2024a) integrates multiple performance-boosting components into TD3+BC and has achieved SOTA results across a wide range of datasets. TD3+BC, wPC, A2PR, and ASPC are all set as the single hyperparameter set, whereas IQL and ReBRAC rely on dataset-specific hyperparameters found via grid search. We reproduce results for TD3+BC, wPC and A2PR. IQL and ReBRAC results are taken from Tarasov et al. (2024a;b). Complete experimental details for each algorithm are provided in the appendix B.2.

The performance comparison is summarized in Table 1. ASPC achieves the best performance on MuJoCo and Maze2d, and exhibits competitive results on Adroit and AntMaze. Most notably, ASPC attains SOTA performance on average across all four domains, which not only outperforms other adaptive policy constraint methods but also surpasses approaches that rely on meticulous per-dataset hyperparameter tuning, highlighting its remarkable adaptability. Figure 2 shows that the performance profile curves (left) place ASPC above all baselines for almost every threshold, and the min-max-scaled radar chart (right) gives ASPC the largest, most balanced polygon, visually confirming its strong and stable performance across tasks without per-dataset tuning.

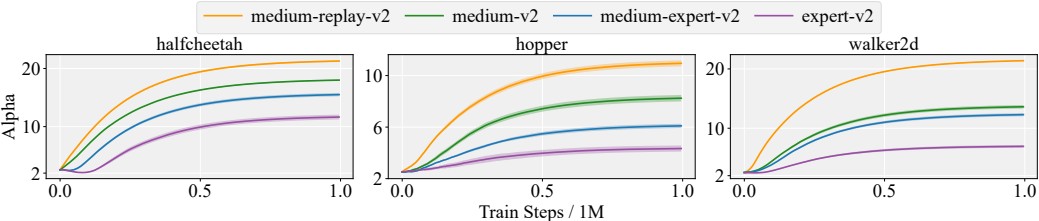

Figure 3: Learning curves of $\alpha$ on halfcheetah, hopper, and walker2d across datasets of different quality. Higher-quality datasets yield smaller $\alpha$ (favoring BC), while lower-quality ones yield larger $\alpha$ (favoring RL). $\alpha$ is initialized to 2.5.

## 5.2 ADAPTABILITY OF THE SACLE FACTOR

**Dataset Adaptability** Figure 3 shows the evolution of $\alpha$ on HalfCheetah, Hopper, and Walker2d for four dataset quality levels, listed from highest to lowest as expert, medium-expert, medium, and medium-replay. Across all three tasks, higher-quality datasets lead to smaller $\alpha$, which places more weight on BC, whereas lower-quality datasets lead to larger $\alpha$, shifting the emphasis toward RL. The consistent ordering confirms that ASPC automatically adjusts the policy-constraint scale to dataset quality without any per-dataset hyperparameter tuning.

**Task Adaptability** Figure 4 plots the $\alpha$ trajectories on six heterogeneous tasks. Tasks such as door, pen, hammer, and relocate possess narrow expert data distributions; here $\alpha$ settles near $10^{-1}$, giving greater weight to BC. Conversely, antmaze and maze2d, whose datasets contain highly sub-optimal trajectories, drive $\alpha$ above 10, shifting emphasis to RL. This task-aware scaling requires no manual tuning and highlights ASPC's cross-task adaptability.

**Training Adaptability** Combining the curves from Figures 4 and 3, we observe a common learning dynamic: $\alpha$ first drops (or rises only slightly) during the early training phase, indicating greater reliance on BC when the policy is still immature. As learning progresses and the critic stabilises, $\alpha$ gradually increases, handing more control to RL. This smooth, stage-wise adjustment underpins ASPC's stable convergence across tasks and datasets.

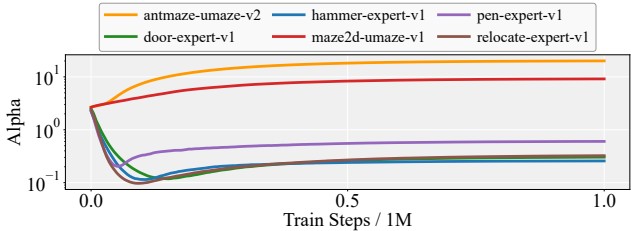

Figure 4: Learning curves of $\alpha$ on six different tasks. The algorithm automatically adjusts $\alpha$ based on the task characteristics. The y-axis is shown in logarithmic scale for better visualization.

## 5.3 NECESSITY OF DYNAMIC SCALE FACTOR ADJUSTMENT

As shown in Table 1, the hyperparameters meticulously selected via grid search ultimately underperform compared to the ASPC algorithm, which dynamically adjusts hyperparameters during training. This observation raises the question: is grid search simply failing to find the best setting, or is the dynamic adjustment in ASPC the true source of its advantage? To answer this, we conduct three controlled tests. **Naive $\alpha$.** TD3+BC is run with a fixed scale factor $\alpha = 2.5$. **Converged $\alpha$.** TD3+BC is run with $\alpha$ fixed to the final value reached by ASPC on the same dataset. **Linear $\alpha$.** TD3+BC starts from $\alpha = 2.5$ and linearly interpolates to the above converged value over the training horizon. To ensure fairness, all TD3+BC variants utilize the same robust critic architecture as ASPC, comprising three hidden layers, each followed by a LayerNorm.

Table 2: Results under different $\alpha$ settings. Values in parentheses indicate the percent difference from Naive. Blue denotes improvement, and red denotes degradation.

| Domain | Naive $\alpha$ | Converged $\alpha$ | Linear $\alpha$ | Dynamic $\alpha$ (ASPC) |
|---|---|---|---|---|
| Mujoco | 70.3 | 79.3 (↑12.8%) | 77.0 (↑9.5%) | 82.1 (↑16.8%) |
| Maze2d | 61.9 | 133.2 (↑115.2%) | 103.3 (↑66.9%) | 147.2 (↑137.8%) |
| AntMaze | 28.7 | 64.1 (↑123.3%) | 56.3 (↑96.2%) | 74.5 (↑159.2%) |
| Adroit | 49.9 | 49.1 (↓1.6%) | 47.6 (↓4.6%) | 55.7 (↑11.6%) |
| Total Avg | 57.0 | 71.8 (↑25.9%) | 66.7 (↑17.0%) | 77.9 (↑36.6%) |

Table 2 summarises the mean normalised scores in the four D4RL domains. Percentages in blue report the relative gain over the naive baseline that fixes $\alpha = 2.5$. Converged $\alpha$ and Linear $\alpha$ both outperform the naive setting, which confirms that the value to which ASPC eventually converges is a much more appropriate scale for the policy constraint. ASPC (Dynamic $\alpha$) still exceeds the Converged variant by a wide margin, and the Linear schedule closes only part of the gap. These results show that simply finding a good fixed $\alpha$ is not enough. Adapting the scale throughout training is essential for the best performance. ASPC provides this dynamic adjustment automatically and therefore achieves the highest overall score.

## 5.4 RUNTIME ANALYSIS

ASPC employs second-order gradient computations for updating $\alpha$, which increases cost. However, its update interval ($k_\alpha$) can be set substantially longer than that of the actor ($k_\pi$), thereby minimizing the additional computational overhead. To evaluate runtime efficiency, we compare the execution time of one million iterations of ASPC against that of other baseline algorithms. Figure 5 presents a bar chart comparing the runtime of ASPC against TD3+BC, CQL, IQL, wPC and A2PR on the halfcheetah-medium-v2 dataset. The results indicate that ASPC introduces only a minimal additional computational overhead beyond that of TD3+BC.

We further analyze the relationship between $k_\alpha$, runtime, and performance, as illustrated in Figure 5. The baseline setting for $k_\alpha$ is 10, we observe that reducing $k_\alpha$ does not lead to significant performance degradation. This suggests that ASPC effectively captures the correct gradient optimization direction, maintaining robustness even when the gradient step size is large. When $k_\alpha$ is set to 30, the runtime

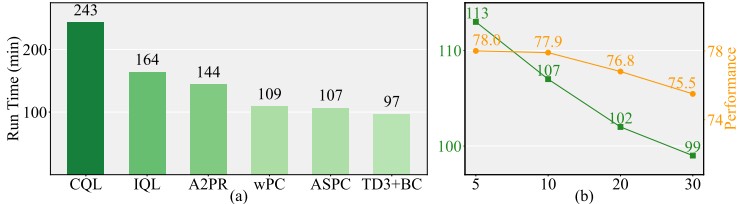

Figure 5: (a) Runtime comparison of different algorithms. (b) Runtime and average performance under different $\alpha$-update intervals ($k_\alpha$). ASPC introduces only minimal overhead compared to TD3+BC, and increasing the update interval reduces runtime while maintaining high performance.

is nearly identical to that of TD3+BC while maintaining strong performance. This highlights the efficiency of the ASPC algorithm.

## 5.5 ABLATION STUDIES

**Robust Critic(RC)** When using the original TD3+BC critic network (with two hidden layers and no LayerNorm), during the process of adjusting $\alpha$, Q-values exhibit significant instability, frequently leading to overestimation, causing catastrophic failure of the algorithm. Since wPC is also designed based on the original TD3+BC framework, we include it in our experiments related to RC (with three hidden layers and LayerNorm). Figure 6a presents the experimental results. The results indicate that when RC is not utilized, both wPC and ASPC achieve limited performance improvement and even exhibit performance degradation on certain tasks.

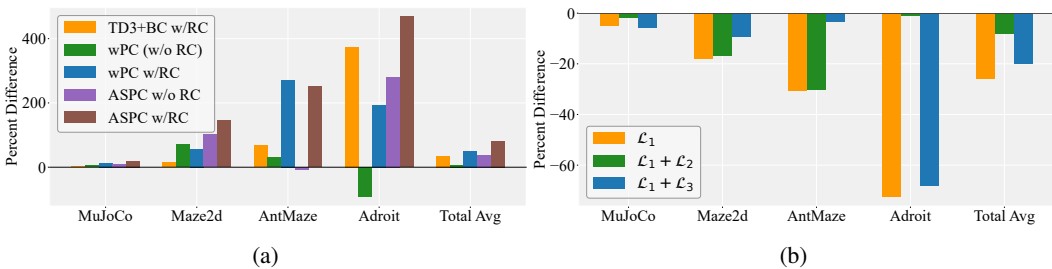

Figure 6: (a) Percent difference relative to the baseline TD3+BC (w/o RC (critic with three hidden layers, each incorporating LayerNorm)). (b) Percent difference of outer loss variants equation 9 relative to the full ASPC configuration.

**Loss Function** Figure 6b reveals clear, domain-dependent effects when the regularization terms are added to the base loss $\mathcal{L}_1$. Adding neither term ($\mathcal{L}_1$ only) gives the poorest performance. Introducing only $\mathcal{L}_2$ lifts performance in MuJoCo and Adroit to the level of full ASPC, while leaving AntMaze almost unchanged. Conversely, adding only $\mathcal{L}_3$ significantly boosts AntMaze but has little effect on MuJoCo or Adroit. For Maze2D, neither single term suffices. Only the full loss $\mathcal{L}_1 + \mathcal{L}_2 + \mathcal{L}_3$ attains the best result. These results can be explained by Proposition 4.2, which shows that $\mathcal{L}_2$ and $\mathcal{L}_3$ implicitly constrain one another. Consequently, adding $\mathcal{L}_2$ in MuJoCo and Adroit implicitly bounds $\Delta L_{BC}$ as well, so the single-step performance guarantee of Theorem 4.4 is already satisfied. Conversely, in AntMaze a direct $\mathcal{L}_3$ penalty implicitly limits $(\Delta Q)^2$, again meeting the theorem's lower bound. For Maze2D, however, neither implicit relation is strong enough; both $\mathcal{L}_2$ and $\mathcal{L}_3$ must be enforced explicitly for the condition in Theorem 4.4 to hold.

## 5.6 EXTENDING ASPC TO OTHER OFFLINE RL METHODS

Many offline RL algorithms follow the form of equation 1. To evaluate the generality of ASPC, we integrate its adaptive policy constraint into three representative baselines, including IQL, CQL, and Diffusion-QL Wang et al. (2023). Each method contains a hyperparameter analogous to $\alpha$ that controls the balance between value learning and conservatism. We replace this manually tuned coefficient with a learnable parameter and update it using the same bi-level second-order procedure as ASPC. The detailed objectives for each algorithm are provided in Appendix D.

Table 3: Performance on Gym-MuJoCo datasets. +ASPC denotes the baseline combined with ASPC, and the percent change indicates its relative improvement over the baseline.

| Gym-MuJoCo | IQL | +ASPC | CQL | +ASPC | Diffusion-QL | +ASPC |
|---|---|---|---|---|---|---|
| halfcheetah-medium | 50.0 | 48.4 (↓3.2%) | 46.8 | 56.3 (↑20.3%) | 51.5 | 59.2 (↑15.0%) |
| halfcheetah-medium-expert | 92.7 | 94.4 (↑1.8%) | 94.2 | 93.6 (↓0.6%) | 96.8 | 96.7 (↓0.1%) |
| halfcheetah-medium-replay | 42.1 | 44.4 (↑5.5%) | 45.3 | 51.0 (↑12.6%) | 47.8 | 58.2 (↑21.8%) |
| hopper-medium | 65.2 | 61.4 (↓5.8%) | 61.3 | 71.6 (↑16.8%) | 90.5 | 101.0 (↑11.6%) |
| hopper-medium-expert | 85.5 | 100.2 (↑17.2%) | 90.1 | 106.9 (↑18.6%) | 111.1 | 111.1 (↑0.0%) |
| hopper-medium-replay | 89.6 | 88.3 (↓1.4%) | 77.5 | 79.9 (↑3.1%) | 101.3 | 100.4 (↓0.9%) |
| walker2d-medium | 80.7 | 83.9 (↑4.0%) | 82.6 | 83.8 (↑1.5%) | 87.0 | 80.3 (↓7.7%) |
| walker2d-medium-expert | 112.1 | 112.1 (↑0.0%) | 109.1 | 109.7 (↑0.6%) | 110.1 | 110.5 (↑0.4%) |
| walker2d-medium-replay | 75.4 | 77.5 (↑2.8%) | 74.5 | 81.7 (↑9.7%) | 95.5 | 95.2 (↓0.3%) |
| **Average** | 77.0 | 79.0 (↑2.5%) | 75.7 | 81.6 (↑7.8%) | 88.0 | 90.3 (↑2.6%) |

As shown in Table 3, incorporating ASPC consistently improves the performance of all three baselines, which demonstrates the broad applicability of our approach. IQL yields the smallest improvement, and a possible reason is that it performs implicit Q learning, so increasing $\alpha$ does not effectively shift the policy toward the RL objective. This implicit structure offers stability but limits the best achievable performance. CQL benefits more from ASPC because updating $\alpha$ directly adjusts the level of conservatism. Diffusion-QL already achieves very strong results, and ASPC further improves its performance, which highlights the robustness of ASPC even when applied to a strong baseline.

## 5.7 ADDITIONAL EXPERIMENTS ON OGBENCH

We further evaluate the generality and robustness of ASPC on OGBench Park et al. (2025a), a new benchmark for offline goal-conditioned RL. Results across ten datasets in Table 4 show that ASPC clearly surpasses all existing baselines, indicating strong applicability beyond D4RL. Since FQL Park et al. (2025b) also follows equation 1, we integrate ASPC by making its scale factor learnable and applying the same bi level optimization procedure, with details in Appendix D. This modification consistently improves FQL, further supporting the broad generality of ASPC across standard and goal-conditioned offline RL.

Table 4: Performance on OGBench. Each entry shows mean $\pm$ std. FQL+ASPC includes the relative performance change over FQL. Bold numbers indicate the best performance for each task.

| OGBench | TD3+BC | IQL | ReBRAC | ASPC | FQL | FQL+ASPC |
|---|---|---|---|---|---|---|
| antmaze-large-navigate-singletask-task1-v0 | 20 ± 44 | 48 ± 9 | 91 ± 10 | **93 ± 4** | 80 ± 8 | 84 (↑5.0%) |
| antmaze-large-navigate-singletask-task2-v0 | 20 ± 31 | 42 ± 6 | **88 ± 4** | 87 ± 7 | 57 ± 10 | 63 (↑10.5%) |
| antmaze-large-navigate-singletask-task3-v0 | 58 ± 31 | 72 ± 7 | 51 ± 18 | **96 ± 4** | 93 ± 3 | 88 (↓5.4%) |
| antmaze-large-navigate-singletask-task4-v0 | 31 ± 37 | 51 ± 9 | 84 ± 7 | **86 ± 5** | 80 ± 4 | 70 (↓12.5%) |
| antmaze-large-navigate-singletask-task5-v0 | 35 ± 38 | 54 ± 2 | **90 ± 2** | 88 ± 4 | 83 ± 4 | 80 (↓3.6%) |
| antmaze-giant-navigate-singletask-task1-v0 | 0 ± 1 | 0 ± 0 | **27 ± 22** | 22 ± 20 | 4 ± 5 | 2 (↓50.00%) |
| antmaze-giant-navigate-singletask-task2-v0 | 15 ± 24 | 1 ± 1 | 16 ± 17 | **74 ± 19** | 9 ± 7 | 26 (↑188.9%) |
| antmaze-giant-navigate-singletask-task3-v0 | 0 ± 1 | 0 ± 0 | **34 ± 22** | 18 ± 13 | 0 ± 1 | 0 (↑0.0%) |
| antmaze-giant-navigate-singletask-task4-v0 | 11 ± 18 | 0 ± 0 | 5 ± 12 | **65 ± 18** | 14 ± 23 | 33 (↑135.7%) |
| antmaze-giant-navigate-singletask-task5-v0 | 16 ± 25 | 19 ± 7 | 49 ± 22 | **55 ± 14** | 16 ± 28 | 49 (↑206.3%) |
| **Average** | 20.6 | 28.7 | 53.5 | **68.4** | 43.6 | 49.5 (↑13.5%) |

## 6 CONCLUSION

We presented ASPC, a bi-level framework that adapts the RL–BC trade off by optimizing the scaling factor $\alpha$ through second-order updates. ASPC yields consistent improvements not only on TD3+BC but also when combined with other offline RL baselines, demonstrating strong generality. However, these simple integrations yield smaller gains than those seen with TD3+BC, indicating that different algorithms may require ASPC-style components tailored to their training dynamics. Future work includes developing such method-specific adaptive mechanisms under a unified principle and evaluating them on larger benchmarks and real-world datasets.

## ACKNOWLEDGMENTS

This work was supported by the National Science and Technology Major Project of the Ministry of Science and Technology of China (2024ZD0608100), the National Natural Science Foundation of China (62506031, 62332017, U22A2022), the Postdoctoral Fellowship Program of CPSF under Grant Number GZC20251093.

## ETHICS STATEMENT

This work focuses on methodological advances in offline RL. All experiments are conducted on standard simulated benchmarks, which do not involve human subjects, personally identifiable information, or sensitive data. We strictly follow the licensing terms of all datasets and simulation platforms used in this study. Our method, Adaptive Scaling of Policy Constraints (ASPC), is designed to improve the stability and reliability of offline RL algorithms. While RL has the potential for deployment in safety-critical domains, such as robotics and autonomous systems, the experiments in this paper remain purely in simulation. Any real-world use of these methods should be preceded by domain-specific safety checks and human oversight to avoid unintended harm.

## REPRODUCIBILITY STATEMENT

We have taken several measures to ensure the reproducibility of our work. The proposed method is described in detail in Section 4, and the complete theoretical derivations are provided in Appendix A. Experimental settings and hyperparameters are reported in Appendix B. Moreover, we include the full implementation code in the Supplementary Material to facilitate replication of all results.

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

# A    THEORETICAL PROOFS

## A.1    PROOF OF PROPOSITION 4.2

Throughout the argument, we adopt the following shorthand. We index the policies as

$$\pi_t \equiv \pi_\theta, \qquad \pi_{t+1} \equiv \pi_{\tilde\theta}.$$

We write

$$L_t^{\mathrm{BC}} := \mathbb{E}_{(s,a)\sim\mathcal{D}}[\|\pi_t(s) - a\|^2], \quad L_{t+1}^{\mathrm{BC}} := \mathbb{E}_{(s,a)\sim\mathcal{D}}[\|\pi_{t+1}(s) - a\|^2],$$

$$\Delta L_{\mathrm{BC}} := |L_{t+1}^{\mathrm{BC}} - L_t^{\mathrm{BC}}|, \quad c := \sqrt{L_t^{\mathrm{BC}}}, \quad x := \mathbb{E}_{s\sim\mathcal{D}}[\|\pi_{t+1}(s) - \pi_t(s)\|^2].$$

**Lemma A.1** (Reverse triangle inequality). *For all $A, B \in \mathbb{R}$ one has $|A + B| \geq \big||A| - |B|\big|$.*

**Lemma A.2** (Cauchy–Schwarz). *For square–integrable real random variables $X, Y$, $\big|\mathbb{E}[XY]\big| \leq \big(\mathbb{E}[X^2]\big)^{1/2}\big(\mathbb{E}[Y^2]\big)^{1/2}$.*

*Proof.*  The proof proceeds in three steps.

Step 1: A lower bound on $\Delta L_{BC}$. Expand the definition of $\Delta L_{BC}$ and simplify:

$$
\begin{aligned}
\Delta L_{BC} &= \left|\mathbb{E}_s\Big[\big\|\pi_{t+1}(s) - \pi_\beta(s)\big\|^2 - \big\|\pi_t(s) - \pi_\beta(s)\big\|^2\Big]\right|\\[4pt]
&= \left|\mathbb{E}_s\Big[\big(\pi_{t+1}(s) - \pi_\beta(s)\big)^\top\big(\pi_{t+1}(s) - \pi_\beta(s)\big) - \big(\pi_t(s) - \pi_\beta(s)\big)^\top\big(\pi_t(s) - \pi_\beta(s)\big)\Big]\right|\\[4pt]
&= \left|\mathbb{E}_s\Big[\big(\pi_{t+1}(s) - \pi_\beta(s) + \pi_t(s) - \pi_\beta(s)\big)^\top\cdot\big(\pi_{t+1}(s) - \pi_t(s)\big)\Big]\right|\\[4pt]
&= \left|\mathbb{E}_s\Big[\big\|\pi_{t+1}(s) - \pi_t(s)\big\|^2 + 2\big(\pi_t(s) - \pi_\beta(s)\big)^\top\big(\pi_{t+1}(s) - \pi_t(s)\big)\Big]\right|\\[4pt]
&= \left|x + 2\,\mathbb{E}_s\Big[\big(\pi_t(s) - \pi_\beta(s)\big)^\top\big(\pi_{t+1}(s) - \pi_t(s)\big)\Big]\right|\\[4pt]
&\overset{A.1}{\geq} |x| - 2\left|\mathbb{E}_s\Big[\big(\pi_t(s) - \pi_\beta(s)\big)^\top\big(\pi_{t+1}(s) - \pi_t(s)\big)\Big]\right|\\[4pt]
&= x - 2\left|\mathbb{E}_s\Big[\big(\pi_t(s) - \pi_\beta(s)\big)^\top\big(\pi_{t+1}(s) - \pi_t(s)\big)\Big]\right|\\[4pt]
&\overset{A.2}{\geq} x - 2\sqrt{\mathbb{E}_s\big\|\pi_t(s) - \pi_\beta(s)\big\|^2}\cdot\sqrt{\mathbb{E}_s\big\|\pi_{t+1}(s) - \pi_t(s)\big\|^2}\\[4pt]
&= x - 2c\sqrt{x}\,.
\end{aligned}
\tag{14}
$$

Since $\Delta L_{BC} \geq 0$ by definition, combining with equation 14 yields

$$\Delta L_{BC} \;\geq\; \max\big\{\, x - 2c\sqrt{x},\; 0 \,\big\}. \tag{15}$$

Step 2: An upper bound on $(\Delta Q)^2$. Jensen's inequality and the assumption 4.1 yield

$$
\begin{aligned}
(\Delta Q)^2 &= \Big(\mathbb{E}_s\big[Q(s, \pi_{t+1}(s)) - Q(s, \pi_t(s))\big]\Big)^2\\[4pt]
&\leq \mathbb{E}_s\big[(Q(s, \pi_{t+1}(s)) - Q(s, \pi_t(s)))^2\big]\\[4pt]
&\leq L_Q^2\,\mathbb{E}_s\,\|\pi_{t+1}(s) - \pi_t(s)\|^2\\[4pt]
&= L_Q^2\,x
\end{aligned}
\tag{16}
$$

Step 3: Mutual bound. From equation 16 we have

$$x \geq x_{\min} := (\Delta Q)^2/L_Q^2. \tag{17}$$

Since $\Delta L_{BC} \geq \max\{x - 2c\sqrt{x}, 0\}$ from equation 15, we relate this expression to $\Delta Q$ as follows. The function $h(x) = x - 2c\sqrt{x}$ is non-positive on $[0, 4c^2]$ and strictly increasing on $[4c^2, \infty)$. When $|\Delta Q| \leq 2cL_Q$, we have $x_{\min} \leq 4c^2$, and $h(x_{\min})$ is non-positive; thus $\Delta L_{BC} \geq 0 \geq h(x_{\min})$. When $|\Delta Q| > 2cL_Q$, we have $x_{\min} > 4c^2$ and $h(x)$ is increasing for all $x \geq x_{\min}$, which gives $\Delta L_{BC} \geq h(x_{\min})$. Combining the two regimes yields the bound

$$\Delta L_{BC} \geq \max\left\{ 0, \ \frac{(\Delta Q)^2}{L_Q^2} - 2c\frac{|\Delta Q|}{L_Q}\right\}. \tag{18}$$

Similarly, using equation 15 we obtain the following upper bounds for $x$:

$$x \leq \left(c + \sqrt{c^2 + \Delta L_{BC}}\right)^2. \tag{19}$$

Combining equation 16 with equation 19 gives an upper bound on $(\Delta Q)^2$:

$$(\Delta Q)^2 \leq L_Q^2 \left(c + \sqrt{c^2 + \Delta L_{BC}}\right)^2. \tag{20}$$

equation 18 and equation 20 together yield the desired mutual bounds. $\qquad\square$

## A.2 PROOF OF PROPOSITION 4.3

This section analyses conditions under which the one-step performance difference $J(\pi_{t+1}) - J(\pi_t)$ admits a tractable lower bound when training on a fixed offline dataset $D$ collected under behavior policy $\pi_\beta$ (so $D \approx d_{\pi_\beta}$).

**Lemma A.3** (Performance-difference lemma). *For any policies $\pi_1$ and $\pi_2$,*

$$J(\pi_1) - J(\pi_2) = \frac{1}{1-\gamma} \mathbb{E}_{s \sim d_{\pi_1}} \left[ \mathbb{E}_{a \sim \pi_1} Q^{\pi_2}(s, a) - V^{\pi_2}(s) \right]. \tag{21}$$

The proof of Lemma A.3 can be found in Kakade & Langford (2002).

**Lemma A.4.** *Under Assumption 4.1, the total variation distance between the visitation distributions of any policy $\pi$ and the behavior policy $\pi_\beta$ satisfies*

$$\|d_\pi - d_{\pi_\beta}\|_1 = \int_s \big|d_\pi(s) - d_{\pi_\beta}(s)\big| \, \mathrm{d}s \leq C \, L_P \max_{s \in \mathcal{S}} \|\pi(s) - \pi_\beta(s)\|. \tag{22}$$

*where $C > 0$ is a constant.*

The proof of Lemma A.4 can be found in the appendix of Xiong et al. (2022).

**Lemma A.5** (Sup-norm version of equation 19). *Define*

$$x_\infty := \sup_s \|\pi_{t+1}(s) - \pi_t(s)\|^2,$$
$$c_\infty^2 := \sup_s \|\pi_t(s) - \pi_\beta(s)\|^2,$$
$$\Delta L_\infty^{BC} := \sup_s \big|\|\pi_{t+1}(s) - \pi_\beta(s)\|^2 - \|\pi_t(s) - \pi_\beta(s)\|^2\big|.$$

*Then*

$$x_\infty \leq \left(c_\infty + \sqrt{c_\infty^2 + \Delta L_\infty^{BC}}\right)^2. \tag{23}$$

*Proof.* For each $s$, let $\Delta L_{BC}(s) = \|\pi_{t+1}(s) - \pi_\beta(s)\|^2 - \|\pi_t(s) - \pi_\beta(s)\|^2$. Then

$$\begin{aligned}
\|\pi_{t+1}(s) - \pi_t(s)\|^2 &= [(\pi_{t+1}(s) - \pi_\beta(s)) - (\pi_t(s) - \pi_\beta(s))]^2 \\
&\leq \left(\|\pi_{t+1}(s) - \pi_\beta(s)\| + \|\pi_t(s) - \pi_\beta(s)\|\right)^2 \\
&= \left(\sqrt{\|\pi_{t+1}(s) - \pi_\beta(s)\|^2} + \|\pi_t(s) - \pi_\beta(s)\|\right)^2 \\
&= \left(\sqrt{\|\pi_t(s) - \pi_\beta(s)\|^2 + \Delta L_{BC}(s)} + \|\pi_t(s) - \pi_\beta(s)\|\right)^2 \\
&\leq \left(c_\infty + \sqrt{c_\infty^2 + \Delta L_\infty^{BC}}\right)^2.
\end{aligned} \tag{24}$$

Taking the supremum over $s$ gives the stated result:

$$x_\infty = \sup_s \|\pi_{t+1}(s) - \pi_t(s)\|^2 \leq \left(c_\infty + \sqrt{c_\infty^2 + \Delta L_\infty^{BC}}\right)^2. \tag{25}$$

The proof of Lemma A.5 is finished. $\qquad\square$

*Proof.* In our deterministic setting, the conditional action distribution $\pi(\cdot|s)$ for any state $s$ is a Dirac measure concentrated at a single action. Specifically, for $\pi_2$ in Lemma A.3 we have:

$$V^{\pi_2}(s) = \mathbb{E}_{a \sim \pi_2}[Q^{\pi_2}(s, a)] = Q^{\pi_2}(s, \pi_2(s)), \tag{26}$$

Applying Lemma A.3 with $\pi_1 = \pi_{t+1}$ and $\pi_2 = \pi_t$ gives:

$$J(\pi_{t+1}) - J(\pi_t) = \frac{1}{1-\gamma} \mathbb{E}_{s \sim d_{\pi_{t+1}}}\left[Q^{\pi_t}(s, \pi_{t+1}(s)) - Q^{\pi_t}(s, \pi_t(s))\right]. \tag{27}$$

Write the performance–difference identity equation 27 as

$$J(\pi_{t+1}) - J(\pi_t) = \frac{1}{1-\gamma} \mathbb{E}_{s \sim d_{\pi_{t+1}}}\left[Q^{\pi_t}(s, \pi_{t+1}(s)) - Q^{\pi_t}(s, \pi_t(s))\right]$$

$$= \frac{1}{1-\gamma}\left\{\mathbb{E}_{s \sim D}\left[Q^{\pi_t}(s, \pi_{t+1}(s)) - Q^{\pi_t}(s, \pi_t(s))\right]\right.$$
$$\left. + \int \left(d_{\pi_{t+1}}(s) - D(s)\right)\left(Q^{\pi_t}(s, \pi_{t+1}(s)) - Q^{\pi_t}(s, \pi_t(s))\right) ds\right\}$$

$$\geq \frac{1}{1-\gamma}\left\{\underbrace{\mathbb{E}_{s \sim D}\left[Q^{\pi_t}(s, \pi_{t+1}(s)) - Q^{\pi_t}(s, \pi_t(s))\right]}_{\Delta Q}\right.$$
$$\left. - \left|\int \left(d_{\pi_{t+1}}(s) - D(s)\right)\left(Q^{\pi_t}(s, \pi_{t+1}(s)) - Q^{\pi_t}(s, \pi_t(s))\right) ds\right|\right\}$$

$$\geq \frac{1}{1-\gamma}\left\{\Delta Q - \|d_{\pi_{t+1}} - d_{\pi_\beta}\|_1 \cdot \sup_s \left|Q^{\pi_t}(s, \pi_{t+1}(s)) - Q^{\pi_t}(s, \pi_t(s))\right|\right\}$$

$$\overset{A.4}{\geq} \frac{1}{1-\gamma}\left\{\Delta Q - C L_P \max_s \|\pi_{t+1} - \pi_\beta\| \cdot \sup_s \left|Q^{\pi_t}(s, \pi_{t+1}(s)) - Q^{\pi_t}(s, \pi_t(s))\right|\right\}$$

$$\overset{4.1}{\geq} \frac{1}{1-\gamma}\left\{\Delta Q - C L_P L_Q \max_s \|\pi_{t+1} - \pi_\beta\| \cdot \max_s \|\pi_{t+1} - \pi_t\|\right\}$$

$$\geq \frac{1}{1-\gamma}\left\{\Delta Q - C L_P L_Q \left(\max_s \|\pi_{t+1} - \pi_t\| + \max_s \|\pi_t - \pi_\beta\|\right) \max_s \|\pi_{t+1} - \pi_t\|\right\}$$

$$= \frac{1}{1-\gamma}\left\{\Delta Q - C L_P L_Q \left(\sqrt{x_\infty} + c_\infty\right)\sqrt{x_\infty}\right\}$$

$$\overset{A.5}{\geq} \frac{1}{1-\gamma}\left\{\Delta Q - C L_P L_Q \left[\left(c_\infty + \sqrt{c_\infty^2 + \Delta L_\infty^{BC}}\right)^2 + c_\infty\sqrt{c_\infty^2 + \Delta L_\infty^{BC}} + c_\infty^2\right]\right\}$$

$$= \frac{1}{1-\gamma}\left\{\Delta Q - C L_P L_Q \left(3c_\infty^2 + 3c\sqrt{c_\infty^2 + \Delta L_\infty^{BC}} + \Delta L_\infty^{BC}\right)\right\}. \tag{28}$$

Thus, the one–step performance satisfies the lower bound

$$J(\pi_{t+1}) - J(\pi_t) \geq \frac{1}{1-\gamma}\left(\Delta Q - \kappa(3c_\infty^2 + 3c\sqrt{c_\infty^2 + \Delta L_\infty^{BC}} + \Delta L_\infty^{BC})\right),$$
$$\kappa := C L_P L_Q. \tag{29}$$

The proof of Proposition 4.3 is finished. $\qquad\square$

A.3 PROOF OF THEOREM 4.4

We now show how our outer-loss components ensure the performance lower bound equation 29 is maintained.

$\mathcal{L}_1$ equation 6 updates $\alpha$ based on the relative gradients of Q-value and the BC loss. Under the initialization assumption $\nabla_\theta \mathbb{E}[Q] > \nabla_\theta L_{BC}$, so $\mathcal{L}_1$ updates $\alpha$ to favor Q-improvement.

In our algorithm, the two regularizers $\mathcal{L}_2$ and $\mathcal{L}_3$ play complementary roles in guaranteeing safe single-step improvements. Specifically, $\mathcal{L}_2$ in equation 7 penalizes the squared change in the Q-function, $\Delta Q^2$, to prevent overly large and unreliable Q-updates. Due to the bootstrapping error inherent in RL, the single-step Q-value changes can be noisy, and therefore we apply an exponential moving average (EMA) for stabilization. In order to preserve the one-step performance lower bound equation 29, $\mathcal{L}_3$ in equation 8 must impose a matching penalty on the bias term identified in that bound. By choosing $\mathcal{L}_3$ so that its curvature mirrors that of $\mathcal{L}_2$, we ensure the single-step performance guarantee remains non-negative.

*Proof.* We perform a second-order Taylor expansion of $\sqrt{c_\infty^2 + \Delta L_\infty^{BC}}$ around $\Delta L_\infty^{BC} = 0$, assuming $\Delta L_\infty^{BC}/c_\infty^2 \ll 1$, discarding higher-order and constant terms. Substituting into the square and retaining only terms up to $O(\Delta L_\infty^{BC})$ yields:

$$
\begin{aligned}
\mathcal{L}_3 &= \kappa^2 \Big( 3c_\infty^2 + 3c_\infty \sqrt{c_\infty^2 + \Delta L_\infty^{BC}} + \Delta L_\infty^{BC} \Big)^2 \\[4pt]
&= \kappa^2 \Big( 3c_\infty^2 + 3c_\infty \Big( c_\infty + \tfrac{\Delta L_\infty^{BC}}{2c_\infty} - \tfrac{(\Delta L_\infty^{BC})^2}{8c_\infty^3} + O(\Delta L_\infty^{BC^3}) \Big) + \Delta L_\infty^{BC} \Big)^2 \\[4pt]
&= \kappa^2 \Big( 6c_\infty^2 + \tfrac{5}{2}\Delta L_\infty^{BC} - \tfrac{3}{8}\frac{(\Delta L_\infty^{BC})^2}{c_\infty^2} + O(\Delta L_\infty^{BC^3}) \Big)^2 \\[4pt]
&= \kappa^2 \Big( 36\,c_\infty^4 + 30\,c_\infty^2\,\Delta L_\infty^{BC} + O(\Delta L_\infty^{BC^2}) \Big) \\[4pt]
&= 36\,\kappa^2\,c_\infty^4 \;+\; 30\,\kappa^2\,c_\infty^2\,\Delta L_\infty^{BC} \;+\; O(\Delta L_\infty^{BC^2}) \\[4pt]
&\approx 30\,\kappa^2\,c_\infty^2\,\Delta L_\infty^{BC} \\[4pt]
&= wc_\infty^2\,\Delta L_\infty^{BC}, \quad w := 30k^2.
\end{aligned}
\tag{30}
$$

In practice, we scale $\mathcal{L}_3$ by the value of $\mathcal{L}_2$ to match its regularization strength and simply set $w$ to 1:

$$
\mathcal{L}_3 = (\Delta Q)^2\, c_\infty^2\, \Delta L_\infty^{BC}.
\tag{31}
$$

By setting an appropriate $w$, the algorithm can guarantee that:

$$
J(\pi_{t+1}) - J(\pi_t) \geq 0.
\tag{32}
$$

The proof of Theorem 4.4 is finished. □

A.4 PROOF OF THEOREM 4.5

*Proof.* We split the total performance gap into two components:

$$
\begin{aligned}
J(\pi^*) - J(\pi_T) &= \big[J(\pi^*) - J(\pi_0)\big] - \big[J(\pi_1) - J(\pi_0)\big] - \big[J(\pi_2) - J(\pi_1)\big] - \cdots - \big[J(\pi_T) - J(\pi_{T-1})\big] \\[4pt]
&= J(\pi^*) - J(\pi_0) - \sum_{i=0}^{T-1} \big[J(\pi_{i+1}) - J(\pi_i)\big].
\end{aligned}
\tag{33}
$$

We first observe that the behavior-cloning loss

$$
L_t^{BC} = \mathbb{E}_{(s,a)\sim D}\big\|\pi_t(s) - a\big\|^2
\tag{34}
$$

decreases rapidly during early training. Hence there exists a warm-up time $t_0$ such that

$$
L_{t_0}^{BC} \leq \varepsilon_0 \implies \mathbb{E}_{s\sim D}\big\|\pi_{t_0}(s) - \beta(s)\big\| \leq \sqrt{\varepsilon_0},
\tag{35}
$$

and we set

$$\pi_0 := \pi_{t_0} \approx \beta.$$

then

$$J(\pi^*) - J(\pi_0) = \frac{1}{1-\gamma} \Big( \mathbb{E}_{s \sim d_{\pi^*}}[\, r(s)\,] - \mathbb{E}_{s \sim d_{\pi_0}}[\, r(s)\,] \Big)$$

$$= \frac{1}{1-\gamma} \int_s \big( d_{\pi^*}(s) - d_{\pi_0}(s) \big)\, r(s)\, \mathrm{d}s$$

$$\leq \frac{1}{1-\gamma} \int_s \big| d_{\pi^*}(s) - d_{\pi_0}(s) \big|\, R_{\max}\, \mathrm{d}s$$

$$= \frac{R_{\max}}{1-\gamma}\, \| d_{\pi^*} - d_{\pi_0} \|_1 \tag{36}$$

$$\overset{A.4}{\leq} \frac{R_{\max}}{1-\gamma}\, C\, L_P\, \max_s \|\pi^*(s) - \pi_0(s)\|$$

$$\leq \frac{C\, L_P\, R_{\max}}{1-\gamma} \Big( \underbrace{\max_s \|\pi^*(s) - \beta(s)\| + \mathbb{E}_{s \sim D}\|\pi_0(s) - \beta(s)\|}_{\varepsilon_\beta} \Big).$$

$$\leq \frac{C\, L_P\, R_{\max}}{1-\gamma} \big( \varepsilon_\beta + \sqrt{\varepsilon_0} \big).$$

We define

$$\Delta_0 = \frac{C\, L_P\, R_{\max}}{1-\gamma} \big( \varepsilon_\beta + \sqrt{\varepsilon_0} \big) \tag{37}$$

Next, each one-step update $i$ produces the gain equation 32

$$\delta_i = J(\pi_{i+1}) - J(\pi_i) \geq 0. \tag{38}$$

Summing these gains yields the unified bound

$$J(\pi^*) - J(\pi_T) \leq \Delta_0 - \sum_{i=0}^{T-1} \delta_i. \tag{39}$$

With a fixed regularization weight $\alpha$, the sequence $\{\delta_i\}$ tends to decay rapidly toward zero or even become negative. Therefore, static $\alpha$ leaves a large residual gap in equation 39. Our meta-update dynamically adjusts $\alpha$ so that each $\delta_i$ stays bounded below by a positive constant $\delta_{\min} > 0$ over a long horizon. Thus

$$J(\pi^*) - J(\pi_T) \leq \Delta_0 - T\,\delta_{\min}. \tag{40}$$

The proof of Theorem 4.5 is finished. $\qquad\square$

# B  EXPERIMENTAL DETAILS

## B.1  HARDWARE AND SOFTWARE

We use the following hadrward:

1) Intel(R) Xeon(R) Platinum 8352V CPU @ 2.10 GHz
2) NVIDIA GeForce RTX 4090 GPU

We use the following software versions:

1) Python 3.8.10
2) D4RL 1.1
3) MuJoCo 3.2.3

4) Gym 0.23.1

5) mujoco-py 2.1.2.14

6) PyTorch 2.2.2 + CUDA 12.1

7) TorchOpt 0.7.3 Ren* et al. (2023)

## B.2 HYPERPARAMETERS

The network structures and hyperparameter configurations of each algorithm corresponding to Table 1 are as follows.

Table 5: ASPC hyperparameters.

|  | Hyperparameter | Value |
|---|---|---|
| TD3+BC hyperparameters | Optimizer | Adam Kingma (2014) |
|  | Critic learning rate | 3e-4 |
|  | Actor learning rate | 3e-4 |
|  | Mini-batch size | 256 |
|  | Discount factor | 0.99 |
|  | Target update rate | 5e-3 |
|  | Policy noise | 0.2 |
|  | Policy noise clipping | (-0.5, 0.5) |
|  | Policy update frequency | 2 |
| Architecture | Critic hidden dim | 256 |
|  | Critic hidden layers | 3 |
|  | Critic activation function | ReLU |
|  | Critic LayerNorm | True |
|  | Actor hidden dim | 256 |
|  | Actor hidden layers | 2 |
|  | Actor activation function | ReLU |
| ASPC hyperparameters | Initial $\alpha$ | 2.5 |
|  | $\alpha$ learning rate | 2e-3 |
|  | $\alpha$ learning rate decay | Exponential |
|  | $\alpha$ update interval | 10 |
|  | EMA smoothing factor | 0.995 |

Table 6: TD3+BC hyperparameters.

|  | Hyperparameter | Value |
|---|---|---|
| TD3+BC hyperparameters | Optimizer | Adam Kingma (2014) |
|  | Critic learning rate | 3e-4 |
|  | Actor learning rate | 3e-4 |
|  | Mini-batch size | 256 |
|  | Discount factor | 0.99 |
|  | Target update rate | 5e-3 |
|  | Policy noise | 0.2 |
|  | Policy noise clipping | (-0.5, 0.5) |
|  | Policy update frequency | 2 |
|  | $\alpha$ | 2.5 |
| Architecture | Critic hidden dim | 256 |
|  | Critic hidden layers | 3 |
|  | Critic activation function | ReLU |
|  | Critic LayerNorm | True |
|  | Actor hidden dim | 256 |
|  | Actor hidden layers | 2 |
|  | Actor activation function | ReLU |

## C  LEARNING CURVES

### C.1  SCALE FACTOR CURVES

Figure 7 plots the $\alpha$ learning curves for all 39 datasets. The curves show that our algorithm (i) drives $\alpha$ toward distinct optima across tasks and (ii) merely modulates its step size and pace when the dataset

Table 7: wPC hyperparameters.

| | Hyperparameter | Value |
|---|---|---|
| | Optimizer | Adam Kingma (2014) |
| | Critic learning rate | 3e-4 |
| | Actor learning rate | 3e-4 |
| | Value learning rate | 3e-4 |
| wPC | Mini-batch size | 256 |
| hyperparameters | Discount factor | 0.99 |
| | Target update rate | 5e-3 |
| | Policy noise | 0.1 |
| | Policy noise clipping | (-0.5, 0.5) |
| | Policy update frequency | 2 |
| | $\alpha$ | 2.5 |
| | Critic hidden dim | 256 |
| | Critic hidden layers | 3 |
| | Critic activation function | ReLU |
| | Critic LayerNorm | True |
| | Actor hidden dim | 256 |
| Architecture | Actor hidden layers | 2 |
| | Actor activation function | ReLU |
| | Value hidden dim | 256 |
| | Value hidden layers | 2 |
| | Value activation function | ReLU |

Table 8: A2PR hyperparameters.

| | Hyper-parameters | Value |
|---|---|---|
| | Optimizer | Adam Kingma (2014) |
| | Critic learning rate | 3e-4 |
| | Actor learning rate | 3e-4 |
| | Mini-batch size | 256 |
| TD3+BC | Discount factor | 0.99 |
| hyperparameters | Target update rate $\tau$ | 5e-3 |
| | Policy noise | 0.2 |
| | Policy noise clipping | (-0.5, 0.5) |
| | Policy update frequency | 2 |
| | $\alpha$ | 2.5 |
| | Q-Critic hidden dim | 256 |
| | Q-Critic hidden layers | 3 |
| | Q-Critic Activation function | ReLU |
| | V-Critic hidden dim | 256 |
| Architecture | V-Critic hidden layers | 3 |
| | V-Critic Activation function | ReLU |
| | Actor hidden dim | 256 |
| | Actor hidden layers | 2 |
| | Actor Activation function | ReLU |
| A2PR | Normalized state | True |
| hyperparameters | $\epsilon_A$ | 0 |
| | $w_1, w_2$ | 1.0 |

Table 9: IQL hyperparameters.

| | Hyperparameter | Value |
|---|---|---|
| | Optimizer | Adam Kingma (2014) |
| | Critic learning rate | 3e-4 |
| | Actor learning rate | 3e-4 |
| IQL | Value learning rate | 3e-4 |
| hyperparameters | Mini-batch size | 256 |
| | Discount factor | 0.99 |
| | Target update rate | 5e-3 |
| | Learning rate decay | Cosine |
| | Critic hidden dim | 256 |
| | Critic hidden layers | 2 |
| | Critic activation function | ReLU |
| | Actor hidden dim | 256 |
| Architecture | Actor hidden layers | 2 |
| | Actor activation function | ReLU |
| | Value hidden dim | 256 |
| | Value hidden layers | 2 |
| | Value activation function | ReLU |

Table 10: IQL's best hyperparameters used in D4RL benchmark.

| Task Name | $\beta$ | IQL $\tau$ | Deterministic policy |
|---|---|---|---|
| halfcheetah-random | 3.0 | 0.95 | False |
| halfcheetah-medium | 3.0 | 0.95 | False |
| halfcheetah-expert | 6.0 | 0.9 | False |
| halfcheetah-medium-expert | 3.0 | 0.7 | False |
| halfcheetah-medium-replay | 3.0 | 0.95 | False |
| halfcheetah-full-replay | 1.0 | 0.7 | False |
| hopper-random | 1.0 | 0.95 | False |
| hopper-medium | 3.0 | 0.7 | **True** |
| hopper-expert | 3.0 | 0.5 | False |
| hopper-medium-expert | 6.0 | 0.7 | False |
| hopper-medium-replay | 6.0 | 0.7 | **True** |
| hopper-full-replay | 10.0 | 0.9 | False |
| walker2d-random | 0.5 | 0.9 | False |
| walker2d-medium | 6.0 | 0.5 | False |
| walker2d-expert | 6.0 | 0.9 | False |
| walker2d-medium-expert | 1.0 | 0.5 | False |
| walker2d-medium-replay | 0.5 | 0.7 | False |
| walker2d-full-replay | 1.0 | 0.7 | False |
| maze2d-umaze | 3.0 | 0.7 | False |
| maze2d-medium | 3.0 | 0.7 | False |
| maze2d-large | 3.0 | 0.7 | False |
| antmaze-umaze | 10.0 | 0.7 | False |
| antmaze-umaze-diverse | 10.0 | 0.95 | False |
| antmaze-medium-play | 6.0 | 0.9 | False |
| antmaze-medium-diverse | 6.0 | 0.9 | False |
| antmaze-large-play | 10.0 | 0.9 | False |
| antmaze-large-diverse | 6.0 | 0.9 | False |
| pen-human | 1.0 | 0.95 | False |
| pen-cloned | 10.0 | 0.9 | False |
| pen-expert | 10.0 | 0.8 | False |
| door-human | 0.5 | 0.9 | False |
| door-cloned | 6.0 | 0.7 | False |
| door-expert | 0.5 | 0.7 | False |
| hammer-human | 3.0 | 0.9 | False |
| hammer-cloned | 6.0 | 0.7 | False |
| hammer-expert | 0.5 | 0.95 | False |
| relocate-human | 1.0 | 0.95 | False |
| relocate-cloned | 6.0 | 0.9 | False |
| relocate-expert | 10.0 | 0.9 | False |

Table 11: ReBRAC hyperparameters.

| | Hyperparameter | Value |
|---|---|---|
| ReBRAC hyperparameters | Optimizer | Adam Kingma (2014) |
| | Mini-batch size | 1024 on Gym-MuJoCo, 256 on others |
| | Learning rate | 1e-3 on Gym-MuJoCo, 1e-4 on AntMaze |
| | Discount factor $\gamma$ | 0.999 on AntMaze, 0.99 on others |
| | Target update rate $\tau$ | 5e-3 |
| Architecture | Hidden dim (all networks) | 256 |
| | Hidden layers (all networks) | 3 |
| | Activation function | ReLU |
| | Critic LayerNorm | True |

Table 12: ReBRAC's best hyperparameters used in D4RL benchmark.

| Task Name | $\beta_1$ (actor) | $\beta_2$ (critic) |
|---|---|---|
| halfcheetah-random | 0.001 | 0.1 |
| halfcheetah-medium | 0.001 | 0.01 |
| halfcheetah-expert | 0.01 | 0.01 |
| halfcheetah-medium-expert | 0.01 | 0.1 |
| halfcheetah-medium-replay | 0.01 | 0.001 |
| halfcheetah-full-replay | 0.001 | 0.1 |
| hopper-random | 0.001 | 0.01 |
| hopper-medium | 0.01 | 0.001 |
| hopper-expert | 0.1 | 0.001 |
| hopper-medium-expert | 0.1 | 0.01 |
| hopper-medium-replay | 0.05 | 0.5 |
| hopper-full-replay | 0.01 | 0.01 |
| walker2d-random | 0.01 | 0.0 |
| walker2d-medium | 0.05 | 0.1 |
| walker2d-expert | 0.01 | 0.5 |
| walker2d-medium-expert | 0.01 | 0.01 |
| walker2d-medium-replay | 0.05 | 0.01 |
| walker2d-full-replay | 0.01 | 0.01 |
| maze2d-umaze | 0.003 | 0.001 |
| maze2d-medium | 0.003 | 0.001 |
| maze2d-large | 0.003 | 0.001 |
| antmaze-umaze | 0.003 | 0.002 |
| antmaze-umaze-diverse | 0.003 | 0.001 |
| antmaze-medium-play | 0.001 | 0.0005 |
| antmaze-medium-diverse | 0.001 | 0.0 |
| antmaze-large-play | 0.002 | 0.001 |
| antmaze-large-diverse | 0.002 | 0.002 |
| pen-human | 0.1 | 0.5 |
| pen-cloned | 0.05 | 0.5 |
| pen-expert | 0.01 | 0.01 |
| door-human | 0.1 | 0.1 |
| door-cloned | 0.01 | 0.1 |
| door-expert | 0.05 | 0.01 |
| hammer-human | 0.01 | 0.5 |
| hammer-cloned | 0.1 | 0.5 |
| hammer-expert | 0.01 | 0.01 |
| relocate-human | 0.1 | 0.01 |
| relocate-cloned | 0.1 | 0.01 |
| relocate-expert | 0.05 | 0.01 |

quality changes within the same task. This dual behaviour highlights the method's adaptability to both task differences and data-quality variations.

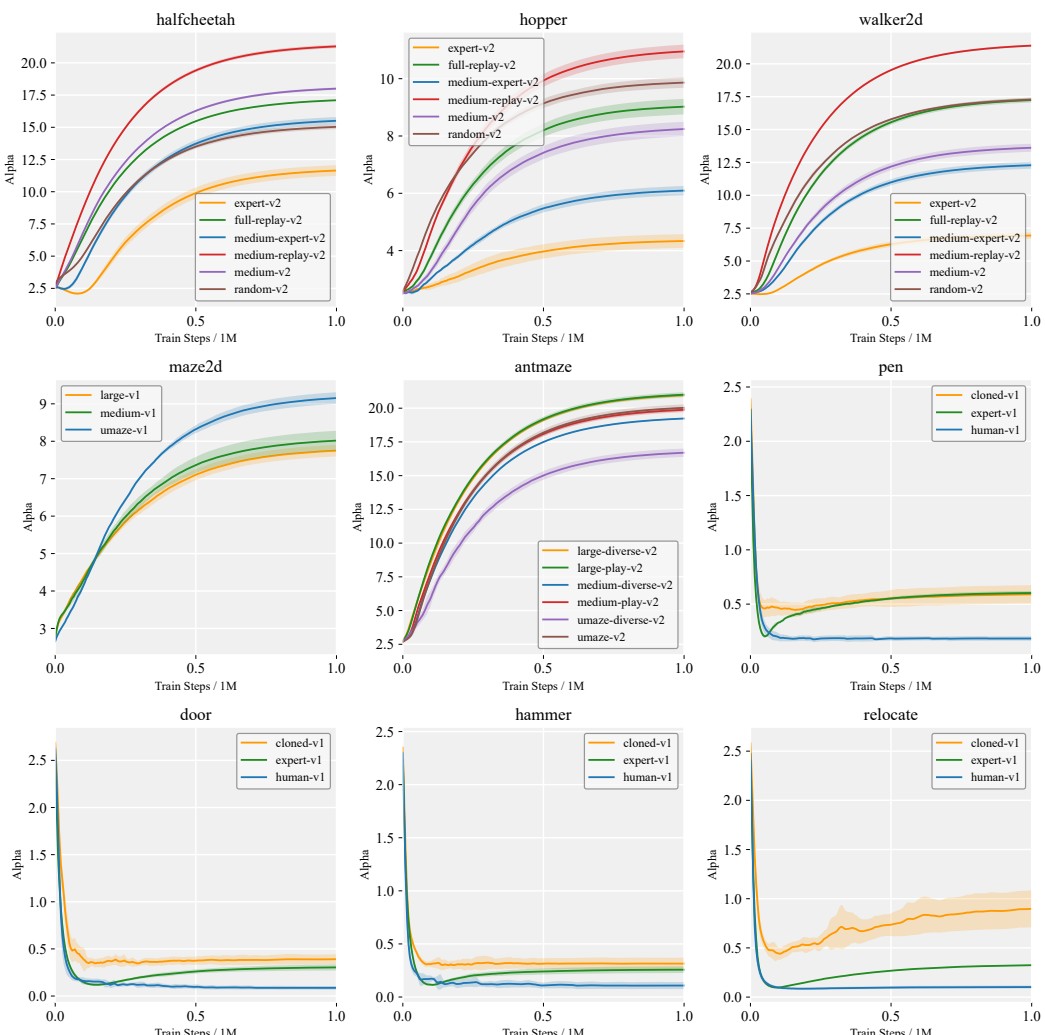

Figure 7: Learning curves of $\alpha$ for nine tasks across 39 datasets.

## C.2 PERFORMANCE CURVES

Figure 8 shows the learning curves of all four algorithms on the 39 D4RL datasets. ASPC rises much more rapidly than the baselines, typically within the first 0.2–0.3 M environment steps, and surpasses them long before the others stabilize. Its final normalized scores are almost always the highest (or very close to the highest) across all task families, maintaining a clear margin where the competing methods usually plateau. Moreover, the shaded regions (mean $\pm$ 1 s.d. over four seeds) remain consistently narrow for ASPC, and its curves show no late-stage collapses, pointing to lower variance and steadier adaptation across widely varying task dynamics and data quality. Overall, the figure suggests that ASPC combines greater sample efficiency, stronger ultimate performance, and more reliable behavior than the other approaches.

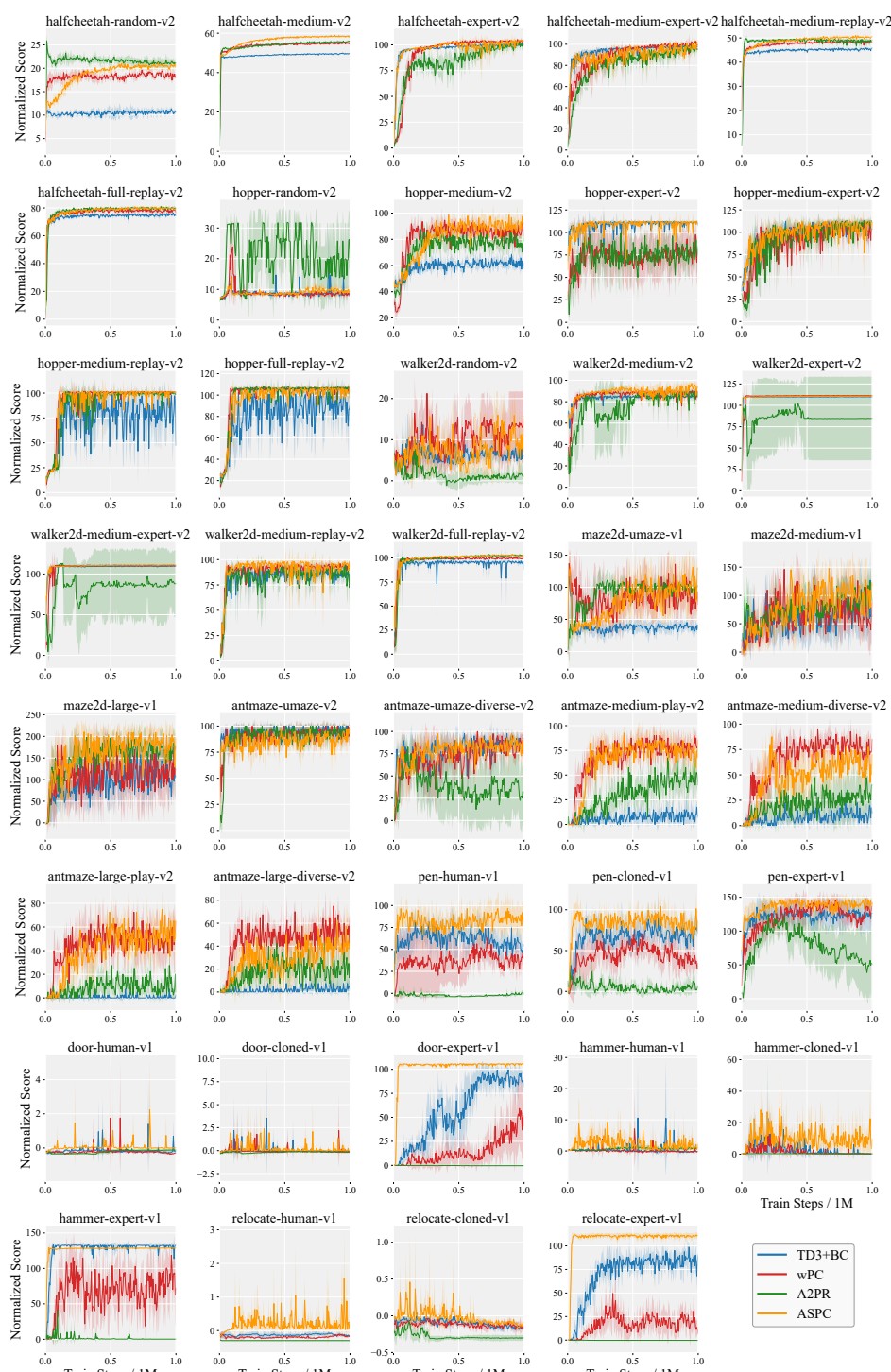

Figure 8: Learning curves comparing the performance of ASPC against other baselines.

## D INTEGRATING ASPC WITH OTHER OFFLINE RL ALGORITHMS

### D.1 INTEGRATION WITH IQL

In IQL, the policy is trained by advantage-weighted behavior cloning. Let $\mathrm{adv}(s, a)$ denote the IQL advantage estimate and $\beta > 0$ the temperature parameter. We integrate ASPC by treating $\beta$ as the adaptive policy-constraint coefficient.

**Inner objective.** The IQL actor minimizes

$$\mathcal{L}_{\mathrm{inner}}^{\mathrm{IQL}}(\theta; \beta) = \mathbb{E}_{(s,a) \sim \mathcal{D}}\Big[\exp(\beta \, \mathrm{adv}(s, a)) \, \ell_{\mathrm{BC}}(\pi_\theta(a \mid s))\Big], \quad (41)$$

where $\ell_{\mathrm{BC}}(\pi_\theta(a \mid s)) = -\log \pi_\theta(a \mid s)$. A single gradient step yields the updated policy $\pi_{\tilde{\theta}(\beta)}$.

**Outer objective.** Following ASPC, we construct an outer loss on the updated policy using a normalized Q-improvement term and the corresponding BC loss:

$$\mathcal{L}_1^{\mathrm{IQL}}(\beta) = -\frac{\mathbb{E}_s[Q(s, \pi_{\tilde{\theta}}(s))]}{\mathbb{E}_s[|Q(s, \pi_{\tilde{\theta}}(s))|]} + \mathbb{E}_{(s,a)}\Big[\exp(\beta \, \mathrm{adv}(s, a)) \, \ell_{\mathrm{BC}}(\pi_{\tilde{\theta}}(a \mid s))\Big]. \quad (42)$$

The second term measures the change in mean Q-value induced by the inner update:

$$\mathcal{L}_2^{\mathrm{IQL}}(\beta) = \Big(\mathbb{E}_s[Q(s, \pi_{\tilde{\theta}}(s))] - \mathbb{E}_s[Q(s, \pi_\theta(s))]\Big)^2. \quad (43)$$

The outer objective for adapting $\beta$ is

$$\mathcal{L}_{\mathrm{outer}}^{\mathrm{IQL}}(\beta) = \mathcal{L}_1^{\mathrm{IQL}}(\beta) + \mathcal{L}_2^{\mathrm{IQL}}(\beta). \quad (44)$$

### D.2 INTEGRATION WITH CQL

CQL constrains Q-values by penalizing larger Q-values on out-of-distribution (OOD) actions. Let $\alpha > 0$ denote the conservatism coefficient. Following ASPC, we treat $\alpha$ as the adaptive policy-constraint parameter.

**Inner objective.** Given a batch $(s, a, r, s')$, the CQL critic update solves

$$\mathcal{L}_{\mathrm{inner}}^{\mathrm{CQL}}(\psi; \alpha) = \underbrace{\mathbb{E}\Big[\big(Q_\psi(s, a) - \mathcal{T}Q(s, a)\big)^2\Big]}_{\text{Bellman regression}} + \alpha \underbrace{\Big(\mathbb{E}_{a' \sim \pi(\cdot \mid s)}[Q_\psi(s, a')] - Q_\psi(s, a)\Big)}_{\text{CQL penalty}}, \quad (45)$$

where

$$\mathcal{T}Q(s, a) = r + \gamma \, \mathbb{E}_{a' \sim \pi(\cdot \mid s')}[\min(Q_{\psi^-}(s', a'))].$$

A single gradient step produces the updated critic $Q_{\tilde{\psi}(\alpha)}$.

**Outer objective.** ASPC evaluates the updated critic with a normalized Q-improvement term and the corresponding CQL penalty, forming

$$\mathcal{L}_1^{\mathrm{CQL}}(\alpha) = -\frac{\mathbb{E}_s[Q_{\tilde{\psi}}(s, \pi(s))]}{\mathbb{E}_s[|Q_{\tilde{\psi}}(s, \pi(s))|]} + \alpha \Big(\mathbb{E}_{a' \sim \pi(\cdot \mid s)}[Q_{\tilde{\psi}}(s, a')] - \mathbb{E}_{(s,a)}[Q_{\tilde{\psi}}(s, a)]\Big). \quad (46)$$

The Q-value change induced by the inner update is

$$\mathcal{L}_2^{\mathrm{CQL}}(\alpha) = \Big(\mathbb{E}_s[Q_{\tilde{\psi}}(s, \pi(s))] - \mathbb{E}_s[Q_\psi(s, \pi(s))]\Big)^2. \quad (47)$$

The outer objective for adapting $\alpha$ becomes

$$\mathcal{L}_{\mathrm{outer}}^{\mathrm{CQL}}(\alpha) = \mathcal{L}_1^{\mathrm{CQL}}(\alpha) + \mathcal{L}_2^{\mathrm{CQL}}(\alpha). \quad (48)$$

### D.3 INTEGRATION WITH DIFFUSION-QL

Diffusion-QL trains a diffusion policy by combining a behavior-cloning loss with a normalized Q-term. Let $\eta > 0$ be the coefficient controlling the trade-off between policy improvement and imitation. Following ASPC, we treat $\eta$ as the adaptive constraint parameter.

**Inner objective.** Given state–action pairs $(s, a)$, the diffusion policy $\pi_\theta$ is trained under the objective

$$\mathcal{L}_{\text{inner}}^{\text{DQL}}(\theta; \eta) = \underbrace{\mathbb{E}_{(s,a) \sim \mathcal{D}}\big[\mathcal{L}_{\text{BC}}(\pi_\theta(s), a)\big]}_{\text{Diffusion behavior cloning}} + \eta \underbrace{\left(-\frac{\mathbb{E}_s\big[Q(s, \pi_\theta(s))\big]}{\mathbb{E}_s\big[|Q(s, \pi_\theta(s))|\big]}\right)}_{\text{normalized Q-improvement}}. \tag{49}$$

A single gradient step produces the updated diffusion policy $\pi_{\tilde{\theta}(\eta)}$.

**Outer objective.** ASPC evaluates the updated diffusion policy through a normalized Q-value term and the corresponding BC term:

$$\mathcal{L}_1^{\text{DQL}}(\eta) = -\eta \frac{\mathbb{E}_s\big[Q(s, \pi_{\tilde{\theta}}(s))\big]}{\mathbb{E}_s\big[|Q(s, \pi_{\tilde{\theta}}(s))|\big]} + \mathbb{E}_{(s,a)}\big[\mathcal{L}_{\text{BC}}\big(\pi_{\tilde{\theta}}(s), a\big)\big]. \tag{50}$$

The Q-improvement induced by the inner update is captured by

$$\mathcal{L}_2^{\text{DQL}}(\eta) = \Big(\mathbb{E}_s[Q(s, \pi_{\tilde{\theta}}(s))] - \mathbb{E}_s[Q(s, \pi_\theta(s))]\Big)^2. \tag{51}$$

The outer objective becomes

$$\mathcal{L}_{\text{outer}}^{\text{DQL}}(\eta) = \mathcal{L}_1^{\text{DQL}}(\eta) + \mathcal{L}_2^{\text{DQL}}(\eta). \tag{52}$$

D.4 INTEGRATION WITH FQL

FQL employs two policies: (i) a teacher flow policy trained purely by flow-matching, and (ii) a student one-step flow policy trained via distillation and Q-improvement. Only the student policy interacts with the Q-function, making it the component that requires adaptive scaling. We integrate ASPC by treating the student's trade-off coefficient $\alpha$ as the adaptive constraint parameter.

**Teacher objective (BC Flow).** The teacher flow policy is trained via standard flow-matching:

$$\mathcal{L}_{\text{teacher}} = \mathbb{E}_{(s,a)}\big[\|f_\theta(s, x_t, t) - (a - x_0)\|^2\big], \tag{53}$$

where $x_t = (1 - t)x_0 + ta$ and $f_\theta$ denotes the flow velocity network. This loss is independent of $\alpha$.

**Inner objective (Student Flow).** The student one-step policy $\pi_\theta$ predicts an action in a single step and matches the teacher via a distillation loss, while also incorporating a normalized Q-improvement term. The inner objective is

$$\mathcal{L}_{\text{inner}}^{\text{FQL}}(\theta; \alpha) = \underbrace{\mathbb{E}_{s, \varepsilon}\big[\|\pi_\theta(s, \varepsilon) - \pi_{\text{teacher}}(s, \varepsilon)\|^2\big]}_{\text{distillation (BC) term}} + \alpha \underbrace{\left(-\frac{\mathbb{E}_s[Q(s, \pi_\theta(s))]}{\mathbb{E}_s\big[|Q(s, \pi_\theta(s))|\big]}\right)}_{\text{normalized Q-improvement}}. \tag{54}$$

A single gradient update produces the updated student policy $\pi_{\tilde{\theta}(\alpha)}$.

**Outer objective.** ASPC evaluates the updated student policy by combining its normalized Q-value and distillation loss, and the Q-improvement incurred by the inner update:

$$\mathcal{L}_1^{\text{FQL}}(\alpha) = -\alpha \frac{\mathbb{E}_s[Q(s, \pi_{\tilde{\theta}}(s))]}{\mathbb{E}_s\big[|Q(s, \pi_{\tilde{\theta}}(s))|\big]} + \mathbb{E}_{s, \varepsilon}\big[\|\pi_{\tilde{\theta}}(s, \varepsilon) - \pi_{\text{teacher}}(s, \varepsilon)\|^2\big], \tag{55}$$

$$\mathcal{L}_2^{\text{FQL}}(\alpha) = \Big(\mathbb{E}_s[Q(s, \pi_{\tilde{\theta}}(s))] - \mathbb{E}_s[Q(s, \pi_\theta(s))]\Big)^2. \tag{56}$$

The outer objective becomes

$$\mathcal{L}_{\text{outer}}^{\text{FQL}}(\alpha) = \mathcal{L}_1^{\text{FQL}}(\alpha) + \mathcal{L}_2^{\text{FQL}}(\alpha). \tag{57}$$

# E    ADDITIONAL EMPIRICAL ANALYSES

## E.1    PERFORMANCE ON ANTMAZE AND ADROIT

As shown in Table 1, ASPC does not achieve state-of-the-art performance on AntMaze and Adroit. Both benchmarks are characterized by extremely sparse rewards, with AntMaze in particular using a binary 0–1 success signal Fu et al. (2020). In such settings, even online RL methods struggle to learn effectively, and behavior cloning plays a dominant role in determining policy quality.

Although ASPC can adapt $\alpha$ toward a more BC-dominated regime, the current BC term imitates all actions in the dataset, including suboptimal or unsuccessful trajectories. This limits the attainable performance on sparse-reward tasks. To address this issue, we experimented with augmenting the BC term using advantage-weighted behavior cloning, where high-advantage samples receive larger weights. The modified loss improves the selectivity of imitation by emphasizing demonstrably good behaviors. Experimental results, shown in Figure 9, indicate consistent performance gains on both AntMaze and Adroit when advantage-weighting is applied. This suggests that selectively imitating high-quality behaviors is crucial for sparse-reward offline RL tasks.

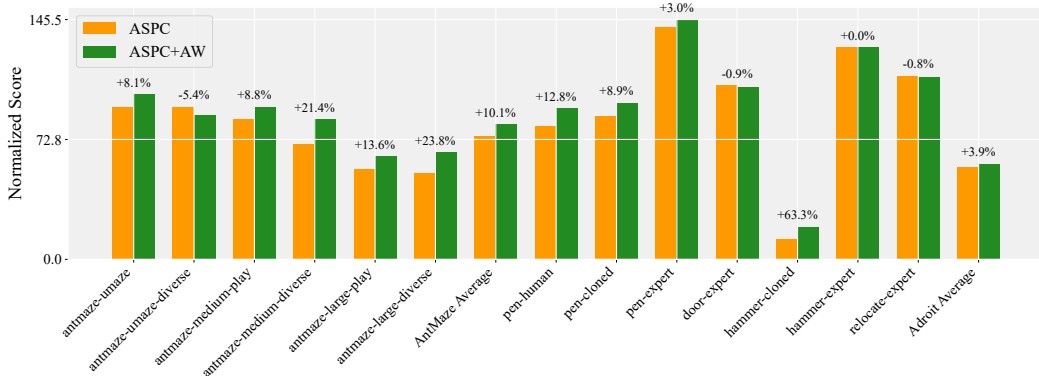

Figure 9: Normalized scores on AntMaze and Adroit tasks. Each pair of bars corresponds to a single dataset (plus the domain-wise average), comparing ASPC (orange) and ASPC+AW (green), where ASPC+AW applies advantage-weighted behavior cloning. The percentages annotated above the green bars indicate the relative performance change of ASPC+AW with respect to ASPC on each task.

## E.2    ABLATION ON THE FORMULATION OF THE $L_3$ TERM

To better understand the role of each component in the $L_3$ term, we consider five variants. The first variant keeps only the third component:

$$L_3^{(1)} = \sup_{(s,a)\in\mathcal{D}} \left| \|\pi_{\tilde{\theta}}(s) - a\|^2 - \|\pi_\theta(s) - a\|^2 \right|.$$

The second variant multiplies the third component by the squared BC deviation:

$$L_3^{(2)} = \left( \sup_{(s,a)\in\mathcal{D}} \|\pi_\theta(s) - a\|^2 \right) L_3^{(1)}.$$

The third variant replaces the BC-deviation factor with the detached $L_2$ term:

$$L_3^{(3)} = (L_2 \text{ detach}) \ L_3^{(1)}.$$

The fourth variant is the complete formulation used in our method:

$$L_3^{(4)} = (L_2 \text{ detach}) \left( \sup_{(s,a)\in\mathcal{D}} \|\pi_\theta(s) - a\|^2 \right) L_3^{(1)}.$$

Table 13: Ablation on different formulations of the $L_3$ term. Values in parentheses denote relative change (%) w.r.t. the full formulation (variant 4). Positive changes are shown in blue, negative in red.

| Formulation | Gym-MuJoCo | Maze2d | AntMaze | Adroit | Total Avg |
|:-:|:-:|:-:|:-:|:-:|:-:|
| (1) | 76.7 (-6.6%) | 97.2 (-34.0%) | 31.7 (-57.4%) | 55.2 (-0.9%) | 64.7 (-16.9%) |
| (2) | 76.8 (-6.5%) | 107.2 (-27.2%) | 31.9 (-57.2%) | 54.9 (-1.4%) | 65.5 (-15.9%) |
| (3) | 81.1 (-1.2%) | 151.8 (+3.1%) | 73.3 (-1.6%) | 56.1 (+0.7%) | 77.7 (-0.2%) |
| (4) | 82.1 | 147.2 | 74.5 | 55.7 | 77.9 |
| (5) | 82.0 (-0.1%) | 149.2 (+1.4%) | 74.6 (+0.1%) | 55.5 (-0.4%) | 77.9 (+0.0%) |

The fifth variant replaces both supremum operators in $L_3^{(4)}$ by dataset expectations:

$$L_3^{(5)} = (L_2 \text{ detach}) \left( \mathbb{E}_{(s,a)\sim\mathcal{D}} \| \pi_\theta(s) - a \|^2 \right) \left| \mathbb{E}_{(s,a)\sim\mathcal{D}} \left[ \| \pi_{\tilde{\theta}}(s) - a \|^2 - \| \pi_\theta(s) - a \|^2 \right] \right|.$$

Table 13 summarizes the results. Variants (3)–(5), which include the detached $L_2$ term, provide clear gains on Maze2d and AntMaze, showing that this component is essential for these domains. By contrast, Adroit displays only minor differences across all variants, suggesting that Q-value gradients dominate BC-related gradients there, making the precise form of $L_3$ less influential. Finally, variant (5) achieves a performance nearly identical to the full formulation, implying that strict worst-case bounds using the sup operator are not essential in practice.

### E.3 CASE STUDY OF ASPC DYNAMICS

Figure 10 shows the training dynamics on halfcheetah-medium-v2. ASPC consistently increases both the estimated Q-value and the BC loss, while simultaneously improving the normalized score. It is essential to note that the increase in BC loss under ASPC does not indicate instability or degradation. Since ASPC deliberately allows the policy to deviate from the behavior policy when such deviations yield sufficient Q-value improvement, the BC loss can increase while performance improves. This matches our theory: whenever the Q-value gain compensates for the increased deviation, the update remains beneficial. Thus, an increasing BC loss indicates that ASPC is escaping the behavior cloning regime and moving toward higher-value actions. In contrast, TD3+BC rapidly plateaus in all three curves, indicating that its fixed trade-off between RL and BC limits its ability to continue improving.

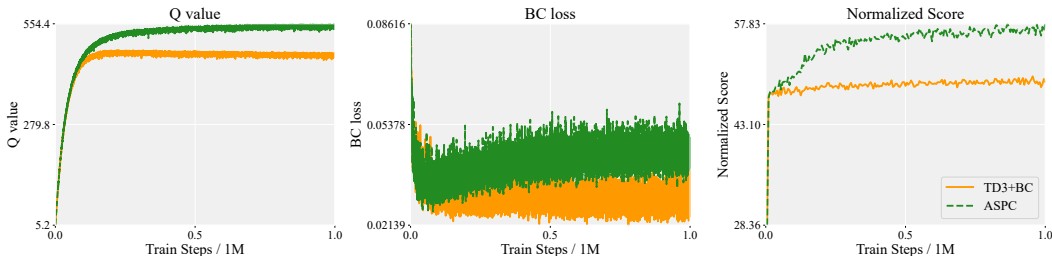

Figure 10: Case study on halfcheetah-medium-v2. ASPC maintains increasing Q-values and BC loss throughout training, accompanied by continuous improvement in normalized score. In contrast, TD3+BC quickly saturates in all three metrics. This behavior is consistent with the theoretical single-step performance improvement condition, illustrating that ASPC sustains stable policy enhancement over the course of training.

## F HYPERPARAMETER SENSITIVITY ANALYSES

### F.1 SENSITIVITY TO THE INITIAL VALUE OF $\alpha$

Figure 11 illustrates the influence of the initial value of $\alpha$ on ASPC. Across Gym-MuJoCo, AntMaze, and Adroit, the intermediate setting $\alpha_0 = 2.5$ provides the strongest overall performance, while a very small initialization ($\alpha_0 = 0.1$) tends to bias the early update dynamics too strongly toward

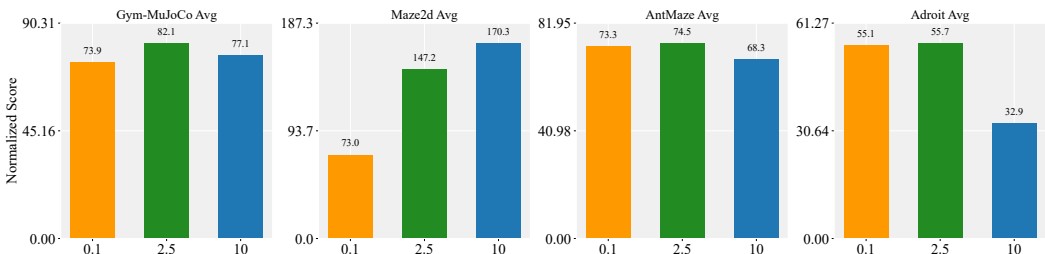

Figure 11: Sensitivity of ASPC to the initial value of $\alpha$. We compare three initializations ($\alpha_0 = 0.1, 2.5, 10$) and report the domain-wise normalized averages.

BC, limiting the contribution of the RL term. Conversely, an excessively large initialization (e.g., $\alpha_0 = 10$) can overemphasize the RL component at the beginning, which weakens the intended stabilizing effect of the BC objective and leads to performance drops, particularly on Adroit. These observations indicate that a balanced initialization is important for achieving stable optimization.

### F.2    SENSITIVITY TO THE LEARNING RATE OF $\alpha$

We study the effect of the learning rate used for updating $\alpha$. The results show that different domains prefer different learning rate magnitudes. Too small values slow down the adjustment of the RL–BC trade-off, while too large values make the meta-update unstable and degrade performance.

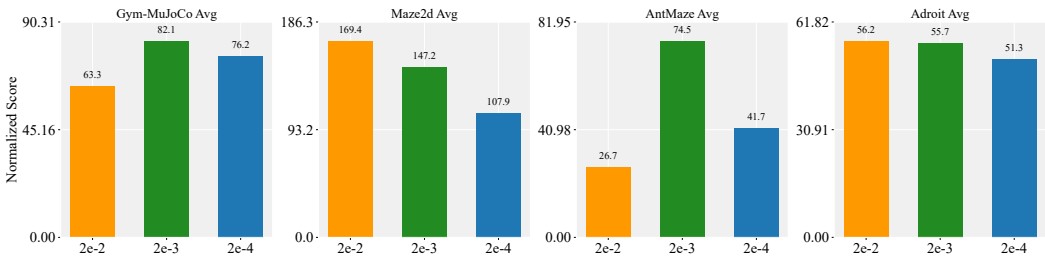

Figure 12: Sensitivity analysis on the learning rate of $\alpha$ across all domains. Each panel reports the domain-level normalized score under three learning rate settings ($2 \times 10^{-2}, 2 \times 10^{-3}, 2 \times 10^{-4}$).

## G    THE USE OF LLM

Large Language Models (LLMs) were used to aid and polish the writing of this paper. In particular, they were applied to rephrase sentences for improved readability and refine grammar and wording to meet academic style requirements.

