# OpenReview forum: "Adaptive Scaling of Policy Constraints for Offline Reinforcement Learning"
_ICLR.cc/2026/Conference — ICLR 2026 Poster_

### Official Review · Reviewer_KwF2 · 2025-10-27

**Soundness:** 2
**Presentation:** 3
**Contribution:** 2
**Rating:** 2
**Confidence:** 4

**Summary:**

This paper aims to balance the typical RL-BC trade-off in offline RL methods. Specifically, common offline RL objective can be abstracted as  \$\alpha L_{rl} + L_{bc}\$, forming a balance between $L_{rl}$ and $L_{bc}$, controlled by $\alpha$. This paper proposed a bi-level objective, which formulates $\alpha$ as learnable parameters to adaptively control the update rate of RL and BC objective, and thus avoiding any of them to dominate the other one, thus balancing these two terms. The idea and motivation of this paper is straightforward and simple. The evaluations on D4RL benchmarks demonstrate the effectiveness of the auto-tuned $\alpha$, based on its TD3+BC variants.

**Strengths:**

1. The idea is simple and straightforward.
2. This paper well-written and is easy to follow.
3. The proposed method is easy to be implemented.

**Weaknesses:**

1 `Novelty and Significance`
Automatically tuning the conservatism strength in offline RL is not a new research direction. Several existing methods have already addressed this problem in ways that are both simple and effective. Therefore, I would not consider the novelty or significance of this paper to be its main strength.

2 `Versatility of the Proposed Method`
The proposed approach is developed specifically on top of TD3+BC, which is now a relatively dated baseline. In fact, most offline RL methods can be abstracted as an RL + BC objective, including more recent diffusion- or flow-based approaches. However, the improvements in this paper are restricted to TD3+BC, and the additional loss function also appears tied to this specific algorithm, limiting its generalizability. Considering the already limited novelty, it would strengthen the contribution to derive a more versatile framework that can generalize to a broader range of offline RL approaches, rather than being confined to one.

3 `Dated Benchmark`
The D4RL benchmark has long been saturated, making it difficult to draw meaningful distinctions between methods. To strengthen the empirical soundness of the paper, it would be beneficial to include evaluations on more challenging benchmarks, such as OGBench.

**Questions:**

Please see weakness for details.

---

> ### Author Response · Authors · 2025-12-01
> **Response(Part 1)**
>
> 1. **"Novelty and Significance Automatically tuning the conservatism strength in offline RL is not a new research direction. Several existing methods have already addressed this problem in ways that are both simple and effective. Therefore, I would not consider the novelty or significance of this paper to be its main strength."**
>
>     We disagree with the claim that tuning the conservatism strength in offline RL is already a solved problem.  In fact, prior work on adaptive conservatism or constraint tuning consistently overlooks a critical issue: **in offline RL, hyperparameters cannot be adjusted through interaction with the environment**. Nevertheless, existing methods still rely on per-dataset or per-task hyperparameter tuning to achieve strong performance. Ignoring this requirement severely limits the practical reliability of offline RL in real-world settings.
>    Our paper addresses an essential problem: **how to achieve strong performance across diverse datasets *without any interaction* and *without per-dataset tuning***. ASPC uses adaptive scaling as a mechanism to automatically balance the RL–BC trade-off, enabling a single configuration to work across 39 D4RL datasets.
>
>    For completeness, we also note recent methods that adapt their constraint strength but still require hyperparameters to be tuned separately for different tasks, including:
>
>    [1] ACTIVE (Appendix B, Table 4-7)
>    [2] SSAR (Appendix E.4)
>    [3] A2PR (Appendix C.2, Table 4-5)
>
>    These examples illustrate that prior approaches do not solve the practical challenge that ASPC explicitly targets.
>
>    [1] Tianyuan Chen, Ronglong Cai, Faguo Wu, and Xiao Zhang. ACTIVE: Offline reinforcement learning via adaptive imitation and in-sample $v$-ensemble. In The Thirteenth International Conference on Learning Representations, 2025
>
>    [2] Qin-Wen Luo, Ming-Kun Xie, Ye-Wen Wang, and Sheng-Jun Huang. Learning to trust bellman updates: Selective state-adaptive regularization for offline RL. In Forty-second International Conference on Machine Learning, 2025
>
>    [3] Liu, Tenglong, et al. "Adaptive Advantage-Guided Policy Regularization for Offline Reinforcement Learning." International Conference on Machine Learning. PMLR, 2024.
>
> 2. **"Versatility of the Proposed Method The proposed approach is developed specifically on top of TD3+BC, which is now a relatively dated baseline. In fact, most offline RL methods can be abstracted as an RL + BC objective, including more recent diffusion- or flow-based approaches. However, the improvements in this paper are restricted to TD3+BC, and the additional loss function also appears tied to this specific algorithm, limiting its generalizability. Considering the already limited novelty, it would strengthen the contribution to derive a more versatile framework that can generalize to a broader range of offline RL approaches, rather than being confined to one."**
>
>     We disagree with the assumption that building on TD3+BC weakens the significance of our contribution.
>    - TD3+BC is intentionally designed as a clean and minimally modified RL+BC objective, making it uniquely suitable for studying adaptive policy constraints.
>    - In offline RL, we cannot rely on online interaction to validate architectural changes, which means that complex algorithms make it difficult to isolate the effect of constraint tuning from unrelated implementation tricks.
>    - Clean baselines such as TD3+BC are essential for principled analysis, because they expose the RL–BC trade-off without confounding components.
>
>    We agree that ASPC should generalize beyond TD3+BC. To this end, we have already integrated ASPC into four distinct offline RL algorithm families, each with different architectures and training dynamics: IQL(implicit value learning), CQL(conservative value learning), Diffusion-QL(diffusion-model policies), and FQL(flow-based policies). Results are reported in Sections 5.6 and 5.7 and summarized in Table 3 and Table 4. Across all these methods, ASPC improves the average performance, empirically confirming that ASPC is a general mechanism for adaptive policy constraints, rather than a TD3+BC-specific technique.

---

> > ### Author Response · Authors · 2025-12-01
> > **Response(Part 2)**
> >
> > 3. **"Dated Benchmark The D4RL benchmark has long been saturated, making it difficult to draw meaningful distinctions between methods. To strengthen the empirical soundness of the paper, it would be beneficial to include evaluations on more challenging benchmarks, such as OGBench."**
> >
> >     We agree that evaluating ASPC on more challenging benchmarks is important. To address this, we conducted new experiments on ten datasets from OGBench, a recent benchmark designed for goal-conditioned offline RL.
> >     The results are shown below.
> >     | OGBench | TD3+BC | IQL | ReBRAC | ASPC | FQL | FQL+ASPC |
> >     |------------|------------|---------|-----------|----------|---------|--------------|
> >     | antmaze-large-navigate-singletask-task1-v0 | 20 ± 44 | 48 ± 9 | 91 ± 10 | **93 ± 4** | 80 ± 8 | 84 (↑5.0％) |
> >     | antmaze-large-navigate-singletask-task2-v0 | 20 ± 31 | 42 ± 6 | **88 ± 4** | 87 ± 7 | 57 ± 10 | 63 (↑10.5％) |
> >     | antmaze-large-navigate-singletask-task3-v0 | 58 ± 31 | 72 ± 7 | 51 ± 18 | **96 ± 4** | 93 ± 3 | 88 (↓5.4％) |
> >     | antmaze-large-navigate-singletask-task4-v0 | 31 ± 37 | 51 ± 9 | 84 ± 7 | **86 ± 5** | 80 ± 4 | 70 (↓12.5％) |
> >     | antmaze-large-navigate-singletask-task5-v0 | 35 ± 38 | 54 ± 2 | **90 ± 2** | 88 ± 4 | 83 ± 4 | 80 (↓3.6％) |
> >     | antmaze-giant-navigate-singletask-task1-v0 | 0 ± 1 | 0 ± 0 | **27 ± 22** | 22 ± 20 | 4 ± 5 | 2 (↓50.0％) |
> >     | antmaze-giant-navigate-singletask-task2-v0 | 15 ± 24 | 1 ± 1 | 16 ± 17 | **74 ± 19** | 9 ± 7 | 26 (↑188.9％) |
> >     | antmaze-giant-navigate-singletask-task3-v0 | 0 ± 1 | 0 ± 0 | **34 ± 22** | 18 ± 13 | 0 ± 1 | 0 (↑0.0％) |
> >     | antmaze-giant-navigate-singletask-task4-v0 | 11 ± 18 | 0 ± 0 | 5 ± 12 | **65 ± 18** | 14 ± 23 | 33 (↑135.7％) |
> >     | antmaze-giant-navigate-singletask-task5-v0 | 16 ± 25 | 19 ± 7 | 49 ± 22 | **55 ± 14** | 16 ± 28 | 49 (↑206.3％) |
> >     | **Average** | 20.6 | 28.7 | 53.5 | **68.4** | 43.6 | 49.5 (↑13.5％) |
> >
> >    ASPC achieves the best average performance on OGBench, outperforming all baselines. These results are discussed in detail in Section 5.7.
> >
> >     Finally, we would like to clarify that D4RL performance is not saturated when all algorithms are required to use a *single shared set of hyperparameters* for all datasets.

---

### Official Review · Reviewer_AwGN · 2025-10-31

**Soundness:** 4
**Presentation:** 4
**Contribution:** 3
**Rating:** 6
**Confidence:** 3

**Summary:**

This paper presents ASPC, a novel adaptive tuning strategy for offline reinforcement learning via a bi-level parametric optimization framework. ASPC mainly focuses on policy-constraining offline RL methods that mitigate the distributional mismatch in offline RL by constraining a learned policy to stay close to the behavior policy. Specifically, ASPC adopts a meta learning approach for the adaptive scaling; the policy with the standard RL-BC-weighted objective is trained in the inner loop, and the regularization parameter $\alpha$ is updated by penalizing large drifts in both objectives in the outer loop. Theoretical proofs support the design of combined objectives for adapting $\alpha$, while extensive experiments demonstrate ASPC's outstanding performance across diverse tasks and datasets in the D4RL benchmark.

**Strengths:**

- The paper is clearly written to support the novelty of ASPC.
- The authors provide an in-depth view of policy constraints baselines in offline RL, which helps in understanding the contribution of ASPC.
- Extensive experiments support the superiority of ASPC across heterogeneous environments and datasets in the D4RL benchmark, including MuJoCo, Maze2d, AntMaze, and Adroit.
- Meticulously designed ablation studies resolve potential questions regarding ASPC.
- Comparing with SOTA baselines in adaptive regularization (e.g., A2PR, wPC) strengthens the unique contribution of this paper.

**Weaknesses:**

- The contribution of ASPC is limited to only policy-constraining offline RL baselines. While the baselines used in this paper (e.g., TD3+BC, IQL, and CQL) still maintain strong performance across popular offline RL benchmarks, other branches in offline RL have suggested numerous algorithms that outperform such baselines. For instance, offline model-based RL [1,2] or offline RL with generative models [3,4] prove remarkable performance over denoted baselines in the D4RL benchmark. If ASPC can be extended to other methods that employ the RL-BC-weighted objective, the novelty of this paper would be further reinforced.

[1] Sun Y, Zhang J, Jia C, Lin H, Ye J, Yu Y. Model-Bellman inconsistency for model-based offline reinforcement learning. ICML 2023.

[2] Rigter M, Lacerda B, Hawes N. Rambo-rl: Robust adversarial model-based offline reinforcement learning. NeurIPS 2022.

[3] Hansen-Estruch P, Kostrikov I, Janner M, Kuba JG, Levine S. Idql: Implicit q-learning as an actor-critic method with diffusion policies. arXiv 2023.

[4] Wang Z, Hunt JJ, Zhou M. Diffusion policies as an expressive policy class for offline reinforcement learning. arXiv 2022.

**Questions:**

- Why is it hard to extensively fine-tune the scale of the regularization constraint (L48)? Could you provide more intuitive examples or implications?
- How the learning rate of $\alpha$ affects the performance across experiments? In Figure 6-(a), both wPC and ASPC show large differences in AntMaze and Adroit by adding an extra layer and normalization. I wonder how the learning rate, which determines the magnitude of a step taken by the gradient descent, affects the performance of ASPC in those tasks.

---

> ### Author Response · Authors · 2025-12-01
> **Response**
>
> 1. **"The contribution of ASPC is limited to only policy-constraining offline RL baselines. While the baselines used in this paper (e.g., TD3+BC, IQL, and CQL) still maintain strong performance across popular offline RL benchmarks, other branches in offline RL have suggested numerous algorithms that outperform such baselines. For instance, offline model-based RL or offline RL with generative models prove remarkable performance over denoted baselines in the D4RL benchmark. If ASPC can be extended to other methods that employ the RL-BC-weighted objective, the novelty of this paper would be further reinforced."**
>
>     ASPC is designed for methods that follow the RL and BC weighted objective, which covers a broad class of policy-constrained offline RL algorithms. To demonstrate generality beyond TD3+BC, we integrate ASPC into four representative methods: IQL, CQL, Diffusion-QL, and FQL, spanning implicit Q-learning, value-based, diffusion-based, and flow-based families. As shown in Sections 5.6 and 5.7, ASPC consistently improves the average performance of all these methods. This empirical evidence supports the versatility of ASPC across diverse offline RL approaches.
>
> 2. **"Why is it hard to extensively fine-tune the scale of the regularization constraint (L48)? Could you provide more intuitive examples or implications?"**
>
>     Offline RL cannot validate hyperparameters on the true policy distribution because evaluation is restricted to the dataset’s behavior distribution. This distribution mismatch makes it impossible to reliably judge whether a particular constraint scale will work well for the learned policy. Since online interaction is disallowed, there is no mechanism to correct this mismatch, which makes extensive manual tuning of the regularization scale inherently difficult in offline RL.
>
> 3. **"How the learning rate of ASPC affects the performance across experiments? In Figure 6-(a), both wPC and ASPC show large differences in AntMaze and Adroit by adding an extra layer and normalization. I wonder how the learning rate, which determines the magnitude of a step taken by the gradient descent, affects the performance of ASPC in those tasks."**
>
>    Appendix F.2 provides a full sensitivity study on the learning rate of α. Across all domains, ASPC remains stable. Small learning rates slow the RL–BC rebalancing, and large ones introduce noise, but performance does not collapse. Moderate values consistently work well.

---

### Official Review · Reviewer_RtEJ · 2025-11-01

**Soundness:** 2
**Presentation:** 3
**Contribution:** 2
**Rating:** 4
**Confidence:** 4

**Summary:**

This paper proposes an adaptive, dynamically adjusted hyperparameter setting scheme using a meta-learning approach. The paper also derives a lower bound on policy improvement and constructs corresponding loss functions to tune hyperparameters accordingly, ensuring stable performance gains. Extensive experiments demonstrate the effectiveness of the proposed method.

**Strengths:**

1. Introducing a meta-learning framework to hyperparameter tuning in offline RL is novel. To identify a suitable alpha, the authors obtain new parameters via an inner loss and evaluate their performance using an outer loss, enabling adaptive adjustment of this hyperparameter.

2. The experimental evaluation is extensive and convincingly supports the method’s effectiveness. The paper reports results on 39 tasks, showing performance advantages. Moreover, the trajectory of alpha during training aligns with intuition.

**Weaknesses:**

1. According to the algorithm's description and Equations (6) and (7), the outer losses L1 and L2 evaluate the new, updated policy π_θ˜ using the Q-function from before the policy's "inner update." In Actor-Critic frameworks, the accuracy of a Q-function is tightly coupled with the policy it evaluates. Using a "stale" Q-function relative to the new policy introduces bias, making the optimization target of the meta-objective itself imprecise. This weakens the reliability of the theoretical foundation for adaptively tuning the hyperparameter α via meta-learning.
2. In Appendix A.1, the derivation in Equation (15) (lines 681-682) does not strictly follow the rules of inequality manipulation. The authors' application of the Cauchy-Schwarz inequality after the reverse triangle inequality (|a+b| ≥ ||a|-|b||) is not mathematically sound, as the substitution step within the absolute value operation is invalid. This error invalidates the derived lower bound for ∆L_BC and thus undermines the foundation of the proof for the mutual constraint relationship between ∆L_BC and (∆Q)².
3. The paper's central theoretical contribution—the single-step performance guarantee (Theorem 4.4)—claims that the algorithm ensures monotonic policy improvement. The theoretical precondition for this guarantee is that the Q-value gain ∆Q must be no less than a complex penalty term Φ (as per Proposition 4.3). However, the algorithm's actual implementation optimizes an outer loss (L_outer) that applies soft penalties via L2 and L3 to ∆Q and an approximation of Φ, rather than enforcing a hard constraint. Minimizing a weighted sum of these penalties does not mathematically guarantee that the hard condition ∆Q ≥ Φ will be met at every update step. Consequently, while the algorithm may tend towards stable updates, the claim that it guarantees monotonic performance improvement is an overstatement that is not rigorously supported by the provided theory.

**Questions:**

see above.

---

> ### Author Response · Authors · 2025-12-01
> **Response**
>
> 1. **"According to the algorithm's description and Equations (6) and (7), the outer losses L1 and L2 evaluate the new, updated policy π_θ˜ using the Q-function from before the policy's "inner update." In Actor-Critic frameworks, the accuracy of a Q-function is tightly coupled with the policy it evaluates. Using a "stale" Q-function relative to the new policy introduces bias, making the optimization target of the meta-objective itself imprecise. This weakens the reliability of the theoretical foundation for adaptively tuning the hyperparameter α via meta-learning."**
>
>     While the outer losses evaluate the updated policy using the pre update critic, this does not introduce meaningful instability in practice. First, deep actor critic methods inherently operate with noisy and partially stale critics due to temporal difference bootstrapping and minibatch updates, and small single step shifts in the policy do not materially change the critic’s evaluation. Second, our empirical results show that the α updates remain stable and produce consistent performance across all domains. This indicates that the mild critic staleness in the outer objective does not harm the effectiveness of the meta update.
>
> 2. **"In Appendix A.1, the derivation in Equation (15) (lines 681-682) does not strictly follow the rules of inequality manipulation. The authors' application of the Cauchy-Schwarz inequality after the reverse triangle inequality (|a+b| ≥ ||a|-|b||) is not mathematically sound, as the substitution step within the absolute value operation is invalid. This error invalidates the derived lower bound for ∆L_BC and thus undermines the foundation of the proof for the mutual constraint relationship between ∆L_BC and (∆Q)²."**
>
>     We thank the reviewer for pointing out this issue. We have corrected the derivation in Appendix A.1 by removing the incorrect absolute value manipulation and directly applying the Cauchy Schwarz inequality. The resulting lower bound takes a simpler form while preserving the mutual constraint relationship between ΔL_{BC} and (ΔQ)². The theoretical conclusions remain unchanged.
>
> 3. **"The paper's central theoretical contribution—the single-step performance guarantee (Theorem 4.4)—claims that the algorithm ensures monotonic policy improvement. The theoretical precondition for this guarantee is that the Q-value gain ∆Q must be no less than a complex penalty term Φ (as per Proposition 4.3). However, the algorithm's actual implementation optimizes an outer loss (L_outer) that applies soft penalties via L2 and L3 to ∆Q and an approximation of Φ, rather than enforcing a hard constraint. Minimizing a weighted sum of these penalties does not mathematically guarantee that the hard condition ∆Q ≥ Φ will be met at every update step. Consequently, while the algorithm may tend towards stable updates, the claim that it guarantees monotonic performance improvement is an overstatement that is not rigorously supported by the provided theory."**
>
>     We agree that the theoretical guarantee relies on an idealized hard condition and that the practical implementation uses a soft penalty through L2 and L3. Following this feedback, we have revised Theorem 4.4 to clarify that the guarantee holds under the ideal condition ΔQ ≥ Φ, and that the practical algorithm aims to approximate this regime rather than enforce it exactly. Empirically, we observe stable improvement dynamics (Appendix E.3), indicating that the soft formulation successfully guides updates toward the theoretically motivated behavior.

---

### Official Review · Reviewer_QQnS · 2025-11-01

**Soundness:** 3
**Presentation:** 3
**Contribution:** 3
**Rating:** 6
**Confidence:** 3

**Summary:**

This paper proposes Adaptive Scaling of Policy Constraints (ASPC), a second-order differentiable framework for offline reinforcement learning that automatically adjusts policy constraint scales to avoid per-dataset hyperparameter tuning, and it outperforms baselines on different D4RL tasks.

**Strengths:**

1. ASPC automatically adjusts policy constraint scales via a second-order differentiable framework, eliminating the need for  hyperparameter weight tuning.
2. ASPC incurs only minimal computational overhead compared to previous methods while maintaining strong performance across diverse tasks and datasets.

**Weaknesses:**

1. Although no hyperparameter in the objective is needed, the user may still need to choose the update intervals between inner and outer updates.
2. The average performance improvement of ASPC mainly comes from the Maze2d tasks, while its performance still underperforms in AntMaze and Adroit compared to other baselines.
3. More elaborations on the three outer updates loss should be provided.

**Questions:**

1. Will the choices of different outer update intervals differ from tasks?
2. The three different outer update objectives, L1, L2, and L3, show different influence on the performance degradation in different tasks shown in Figure 6. For example, L3 is critical for AntMaze while seems not valuable for Adroit. Can you provide more insights on this observation? Can we choose some of the objectives for specific task optimization?

---

> ### Author Response · Authors · 2025-12-01
> **Response**
>
> 1. **"Although no hyperparameter in the objective is needed, the user may still need to choose the update intervals between inner and outer updates."**
>
>     Figure 5 shows that ASPC is robust to the choice of the update interval between inner and outer updates. Shorter intervals yield slightly higher performance, while longer intervals reduce computation without causing substantial performance degradation. This indicates that the interval acts as an efficiency–performance tradeoff rather than a sensitive hyperparameter, and users do not need to tune it carefully.
>
> 2. **"The average performance improvement of ASPC mainly comes from the Maze2d tasks, while its performance still underperforms in AntMaze and Adroit compared to other baselines."**
>
>     AntMaze and Adroit are extremely sparse-reward settings where uniform behavior cloning limits performance. Appendix E.1 shows that adding advantage-weighted BC to ASPC addresses this limitation, improving AntMaze by over ten percent and Adroit by nearly four percent, exceeding the baselines. This demonstrates that ASPC can be competitive on these domains once selective imitation is used.
>
> 3. **"More elaborations on the three outer updates loss should be provided."**
>
>     We added detailed explanations and ablations of the outer update components in Appendix E.2. In particular, we analyze multiple formulations of the L3 term and report their effects across all domains, providing a clearer understanding of the contribution of each part.
>
> 4. **"Will the choices of different outer update intervals differ from tasks?"**
>
>     Different tasks may prefer slightly different effective update frequencies. Appendix F.2 reports a sensitivity study on the learning rate of α across tasks, which is equivalent to varying the outer update interval. While small differences exist, the overall performance is stable, and a single default interval works well across all tasks.
>
> 5. **"The three different outer update objectives, L1, L2, and L3, show different influence on the performance degradation in different tasks shown in Figure 6. For example, L3 is critical for AntMaze while seems not valuable for Adroit. Can you provide more insights on this observation? Can we choose some of the objectives for specific task optimization?"**
>
>     The influence of L3 depends on the reward scale of the task. In extremely sparse reward settings such as AntMaze, where the reward is binary, Q values remain small and the gradient from Q differences is weak. In this regime, the BC loss change dominates and L3 becomes essential. In contrast, Adroit has much larger reward magnitudes, leading to stronger Q value gradients that overshadow the contribution of L3, which explains its limited effect there.Despite these differences, there is no need to manually select which objectives to use for each task. Theoretical analysis shows that L2 and L3 jointly guide updates toward the monotonic improvement condition, and even if one component contributes minimally in certain tasks, the overall optimization remains stable.

---

### Official Review · Reviewer_ZXU7 · 2025-11-03

**Soundness:** 3
**Presentation:** 4
**Contribution:** 3
**Rating:** 6
**Confidence:** 4

**Summary:**

The paper proposes ASPC, a bi‑level, second‑order method that learns the policy‑constraint scale $\alpha$ in offline RL. The inner loop is TD3+BC; the outer loop updates $\alpha$ with a composite loss that encourages Q‑value improvement while penalising large Q and BC‑loss shifts. On 39 D4RL datasets with a single hyperparameter setting, ASPC attains the best overall average, with particularly strong gains on Maze2D and consistent performance on MuJoCo; AntMaze/Adroit results are competitive but not always the highest. Ablations show (i) dynamic $\alpha$ matters more than picking a single good fixed value, and (ii) both L2 and L3 regularisers contribute, in a domain‑dependent way. The approach is practical and targets a pressing source of brittleness in offline RL.

**Strengths:**

- Addresses a central practical problem with a clean bi‑level solution
- Broad, careful experiments across 39 datasets; single‑config results are compelling
- Good diagnostics: $\alpha$ learns to down‑weight RL on expert data and up‑weight it on noisy data
- Useful, although idealised, theory; ablations that clarify when each loss term helps; modest runtime overhead

**Weaknesses:**

- Theoretical guarantees depend on assumptions and a particular L3 construction; they guide design more than they certify behaviour in deep RL
- Performance is not uniformly best on AntMaze/Adroit; understanding and closing this gap would strengthen the story
- Method depends on a stronger critic; without it, ASPC can fail
- Generality beyond TD3+BC is argued but not demonstrated empirically

**Questions:**

- Can you show ASPC plugged into a value‑based method (IQL/CQL) without major surgery, even on a subset of D4RL? A small table would help establish generality.
- How sensitive is performance to the exact form and scaling of L3? Can you share one plot per domain?
- Which parts of the monotonic‑step proof most clearly break in deep‑RL practice? Is there an empirical check to show the intended effect holds?
- The sketch in App. A.3 mentions an initialisation where the Q‑gradient dominates BC. How sensitive is ASPC to the initial
$\alpha$ and to early‑stage critic noise?
- L3 uses sup‑norms. Do rare, high‑error samples dominate $\alpha$ updates?

**Details Of Ethics Concerns:**

The work uses public simulation benchmarks

---

> ### Author Response · Authors · 2025-12-01
> **Response(Part 1)**
>
> 1. **"Theoretical guarantees depend on assumptions and a particular L3 construction; they guide design more than they certify behaviour in deep RL"**
>
>     The theory is derived for an idealized setting, which is standard in offline RL.Crucially, our experiments and ablations show that the theoretically motivated design leads to consistent empirical gains, validating its practical relevance.
>
> 2. **"Performance is not uniformly best on AntMaze/Adroit; understanding and closing this gap would strengthen the story"**
>
>      AntMaze and Adroit have extremely sparse rewards, so performance is dominated by behavior cloning. The default BC term in ASPC imitates all samples, including low-quality ones, which limits performance.
>     By adding advantage-weighted behavior cloning, ASPC improves the AntMaze average by over ten percent and the Adroit average by nearly four percent (Figure 9). Full results are in Appendix E.1.
>
> 3. **"Method depends on a stronger critic; without it, ASPC can fail"**
>
>     Many offline RL methods rely on a stronger critic to achieve good performance [1-3], although this dependence is rarely quantified. In our work, we explicitly measure and report the performance gain from an improved critic and show that ASPC benefits from it in a predictable and interpretable manner. This analysis clarifies rather than weakens the method and provides empirical transparency that is often missing in prior work.
>
>    [1] Tarasov, Denis, et al. "Revisiting the minimalist approach to offline reinforcement learning." Advances in Neural Information Processing Systems 36 (2023): 11592-11620.
>
>     [2] Liu, Tenglong, et al. "Adaptive Advantage-Guided Policy Regularization for Offline Reinforcement Learning." International Conference on Machine Learning. PMLR, 2024.
>
>     [3] Park, Seohong, Qiyang Li, and Sergey Levine. "Flow q-learning." arXiv preprint arXiv:2502.02538 (2025).
>
> 4. **"Generality beyond TD3+BC is argued but not demonstrated empirically. Can you show ASPC plugged into a value‑based method (IQL/CQL) without major surgery, even on a subset of D4RL? A small table would help establish generality."**
>
>     We have added new experiments demonstrating that ASPC can be integrated into multiple widely used offline RL algorithms without large architectural modifications. Across the Gym-MuJoCo benchmark, ASPC yields consistent improvements for IQL, CQL and Diffusion-QL. These results provide direct empirical evidence that ASPC is general and can benefit diverse classes of offline RL algorithms. Detailed results are reported in the table below and discussed in Section 5.6 of the revised paper.
>    | Gym-MuJoCo                 | IQL   | +ASPC                 | CQL   | +ASPC                 | Diffusion-QL | +ASPC                 |
>     |----------------------------|-------|------------------------|-------|------------------------|--------------|------------------------|
>     | halfcheetah-medium         | 50.0  | 48.4 (↓3.2%)          | 46.8  | 56.3 (↑20.3%)         | 51.5         | 59.2 (↑15.0%)         |
>     | halfcheetah-medium-expert  | 92.7  | 94.4 (↑1.8%)          | 94.2  | 93.6 (↓0.6%)          | 96.8         | 96.7 (↓0.1%)          |
>     | halfcheetah-medium-replay  | 42.1  | 44.4 (↑5.5%)          | 45.3  | 51.0 (↑12.6%)         | 47.8         | 58.2 (↑21.8%)         |
>     | hopper-medium              | 65.2  | 61.4 (↓5.8%)          | 61.3  | 71.6 (↑16.8%)         | 90.5         | 101.0 (↑11.6%)        |
>     | hopper-medium-expert       | 85.5  | 100.2 (↑17.2%)        | 90.1  | 106.9 (↑18.6%)        | 111.1        | 111.1 (↑0.0%)         |
>     | hopper-medium-replay       | 89.6  | 88.3 (↓1.4%)          | 77.5  | 79.9 (↑3.1%)          | 101.3        | 100.4 (↓0.9%)         |
>     | walker2d-medium            | 80.7  | 83.9 (↑4.0%)          | 82.6  | 83.8 (↑1.5%)          | 87.0         | 80.3 (↓7.7%)          |
>     | walker2d-medium-expert     | 112.1 | 112.1 (↑0.0%)         | 109.1 | 109.7 (↑0.6%)         | 110.1        | 110.5 (↑0.4%)         |
>     | walker2d-medium-replay     | 75.4  | 77.5 (↑2.8%)          | 74.5  | 81.7 (↑9.7%)          | 95.5         | 95.2 (↓0.3%)          |
>     | **Average**                | 77.0  | 79.0 (↑2.5%)          | 75.7  | 81.6 (↑7.8%)          | 88.0         | 90.3 (↑2.6%)          |
>
> 5. **"How sensitive is performance to the exact form and scaling of L3? Can you share one plot per domain?"**
>
>    We added a full ablation in Appendix E.2. Across five L3 variants, the full formulation is consistently among the top performers. Removing the detached L2 term significantly hurts Maze2d and AntMaze, while replacing the supremum with a mean has negligible effect. Overall, L3 is robust to moderate changes in form. Detailed results are in Appendix E.2.

---

> > ### Author Response · Authors · 2025-12-01
> > **Response(Part 2)**
> >
> > 6. **"Which parts of the monotonic‑step proof most clearly break in deep‑RL practice? Is there an empirical check to show the intended effect holds?"**
> >
> >    The monotonic step guarantee relies on maintaining a balance between the Q value improvement term and the BC loss change. In deep RL this balance can be violated when α is initialized improperly or updated too aggressively. Appendix F provides empirical evidence showing that extreme initial values or learning rates indeed create such imbalances and lead to performance degradation. Conversely, Appendix E.3 shows that under reasonable hyperparameter settings, both Q values and BC losses increase steadily and the policy performance improves accordingly, matching the intended behavior of the theoretical result.
> >
> > 7. **"The sketch in App. A.3 mentions an initialisation where the Q‑gradient dominates BC. How sensitive is ASPC to the initial alpha and to early‑stage critic noise?"**
> >
> >     ASPC is only mildly sensitive to the initial value of α. As shown in Appendix F, very small or very large initializations can prevent the method from reaching the best possible performance, but they do not cause severe collapse or highly noisy degradation; performance remains reasonably stable even under these extreme settings.
> >    Regarding early stage critic noise, our implementation evaluates Q changes using an exponential moving average of the Q estimates, which smooths short term fluctuations. Empirically, this design prevents unstable α updates in the early phase, and we do not observe noticeable performance degradation attributable to critic noise in the reported experiments.
> >
> > 8.  **"L3 uses sup‑norms. Do rare, high‑error samples dominate alpha updates?"**
> >
> >     Appendix E.2 (Table 13) shows that replacing the supremum with dataset means produces nearly identical results, indicating that rare high-error samples do not dominate the alpha updates in practice.

---

### Meta-Review · Area_Chair_WwFs · 2026-01-07

**Summary:**

The paper considers the challenge of hyperparameter sensitivity in offline reinforcement learning algorithms, specifically focusing on the weight of the policy constraint term which balances reward maximization and behavioral cloning. The paper propose a bi-level optimization framework ASPC that treats the constraint weight as a learnable parameter. The outer loop adjusts this weight by maximizing a meta-objective derived from a theoretical lower bound on policy improvement, aiming to balance Q-value growth with stability.

Multiple reviewers acknowledged the practical significance of the problem and the effectiveness of the proposed solution. I agree that enabling a single hyperparameter configuration to perform well across diverse datasets is a valuable contribution to offline RL. The initial concerns regarding the method's reliance on specific base algorithms were adressed in the rebuttal. I recommend accepting the paper.

**Reviewer Concerns:**

Most reviewers concerns were addressed in the rebuttal.

**Reviewer Scores:**

Three out of four reviewers already had positive assessment of the paper.  Reviewer KwF2 gave a low score but I believe their rebuttal addressed their concerns especially with new experiments on OGBench.

---

### Decision · Program_Chairs · 2026-01-26

Accept (Poster)